# Partial Fusion of Neural Networks:
# Efficient Tradeoffs Between Ensembles and Weight Aggregation

**Fabian Morelli** [1]  **Stephan Eckstein** [2]

## Abstract

Ensembles of neural networks typically outperform individual networks but incur large computational costs, whereas weight aggregation produces less costly, yet also less accurate, aggregate models. We introduce partial fusion of networks, which interpolates between ensembles and weight aggregation and thus allows for a flexible tradeoff between computational cost and performance. A direct way to achieve this is to extend existing weight aggregation methods based on neuron-level similarity between different networks, where partial fusion then only aggregates weights of neurons which are most similar. We showcase one particular method to jointly identify which neurons are most similar and match them via partial optimal transport. Further, we consider the more general perspective of weight aggregation and partial fusion as generalized pruning of ensemble models, where neurons cannot just be deleted, but also linearly combined. Finally, we show that generalized pruning applied to a single network yields similar benefits as partial fusion by allowing for a tradeoff between isolating, deleting, and linearly combining neurons based on similarity. Our code is available at https://github.com/Fabian-Mor/partial_fusion_nn.

## 1. Introduction

Different methods for combining neural networks have been developed and used with great success. Two popular techniques are weight averaging and ensembles. While ensembles have a long history and lead to an increase in robustness and prediction performance by combining the output of sev-

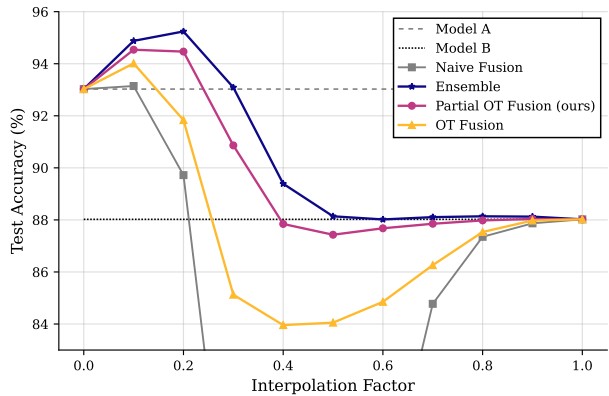

*(a)* Aggregating two MLP models $A$ and $B$ trained on different parts of MNIST. The Interpolation Factor determines the weight given to each of the models in the aggregation. The Partial OT Fusion method is in between weight aggregation (OT Fusion, cf. Singh & Jaggi (2020)) and the ensemble. It yields accuracy closer to the ensemble than to the aggregated model, while only having around $1.45\times$ the amount of parameters of the original networks (compared to $2\times$ for the ensemble).

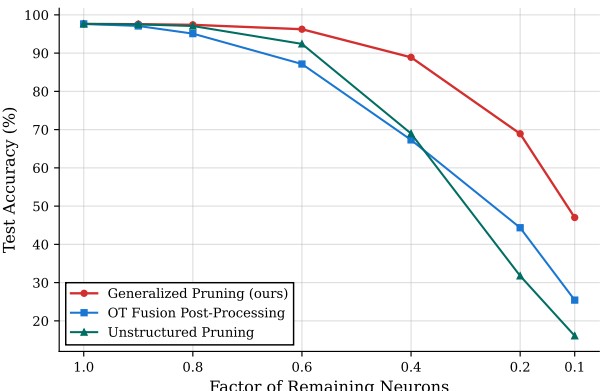

*(b)* The size of a network trained on MNIST is reduced via different methods. Unstructured pruning deletes neurons in each layer and post-processing (cf. Singh & Jaggi (2020)) then merges all initial neurons with the remaining ones. Generalized pruning (with clustering) can leave neurons isolated (as in pruning), but also allows for linearly combining neurons (as in post-processing), thus enabling a similarly flexible tradeoff as the partial fusion method.

*Figure 1.* Illustration of the main ideas in the paper: (a) - model aggregation and (b) - generalized pruning.

[1]Department of Computer Science, University of Tübingen, Germany  [2]Department of Mathematics, University of Tübingen, Germany.  Correspondence to:  Fabian Morelli <research@fabianmorelli.de>.

*Proceedings of the $43^{rd}$ International Conference on Machine Learning*, Seoul, South Korea. PMLR 306, 2026. Copyright 2026 by the author(s).

1. Similarity of neurons via embedding

2. Partial alignment of neurons

3. Partial fusion of neural networks

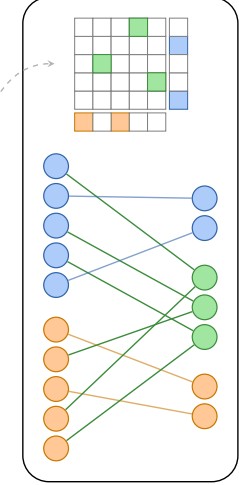
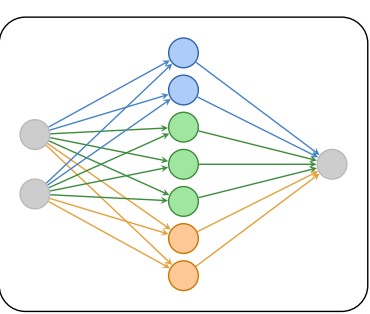

*Figure 2.* Idea behind partial model fusion of two shallow networks. First, based on a feature embedding of the neurons in the hidden layer, the neurons' similarity across the two networks is assessed (left image). Second, the most similar neurons are matched, while the remaining ones are left isolated, leading to a partial alignment-matrix (middle image). In the proposed Partial OT Fusion method, the partial alignment matrix is obtained by solving a partial optimal transport problem. Third, the resulting partially fused network (right image) keeps the initial weights to the isolated neurons, while averaging the weights for the neurons which were matched. The extreme cases are precisely normal weight averaging (if all neurons are matched) and an ensemble model (if no neurons are matched).

eral networks, they come at the cost of keeping several models in memory and increased inference time. Weight averaging (see, e.g., Horoi et al. (2024); Ilharco et al. (2023); Matena & Raffel (2022); McMahan et al. (2017); Wortsman et al. (2022a); Yadav et al. (2023)) is an intriguing alternative, as it offers a way to combine networks without increasing the needed memory and the inference time. Simple weight averaging which does not align neurons before averaging only results in good performing models in limited settings, e.g., in relation to fine-tuning (Wortsman et al., 2022b). Successfully merging neural networks in this simple way is also known as linear mode connectivity (Frankle et al., 2020; Ferbach et al., 2024). Beyond performance improvements, weight aggregation is also an important tool to understand the loss-landscape of neural networks (Akash et al., 2022; Fort & Jastrzebski, 2019; Garipov et al., 2018; Izmailov et al., 2018; Li et al., 2018).

As neural networks are invariant to permutations (Entezari et al., 2022), aligning networks layer-wise via permutations before merging more often allows the merging of networks, even when linear mode connectivity is not given (Ainsworth et al., 2023). A remaining obstacle is that permutation-invariant methods are restricted to merging layers of the same size. In contrast, many recent methods (Singh & Jaggi, 2020; Imfeld et al., 2024; Zhang et al., 2025) also apply to layers of uneven size or even different architectures (Nguyen et al., 2023; Xu et al., 2024b).

In this paper, we propose a novel method to aggregate models which we call partial fusion, that combines the ideas of weight averaging and ensembles. By merging only the most similar parts of the layers of two networks and keeping the rest unchanged, we allow for a flexible layer-wise combination of weight averaging and ensembles.

The core idea in a simple case of two shallow networks is illustrated in Figure 2. While permutation-invariant model fusion aims to match *all* neurons between two layers based on the similarity of their features, partial fusion simply allows for a matching of only the most similar neurons. A computationally tractable way to obtain such a matching of only the most similar neurons is given by partial optimal transport (Figalli, 2010). The resulting aggregate models are then directly in between OT Fusion (Singh & Jaggi, 2020) and ensembles, see Figure 1a. We emphasize that the partial fusion method is however not restricted to one particular way to match (the most similar) neurons, but any way to obtain a partial transformation between layers of different networks leads to a single partially fused neural network, see Section 2.2 for details.

The idea of merging only well-aligned neurons while keeping the rest separate was recently also pursued by Nasery et al. (2025), whose PLeaS method extends Git Re Basin's permutation matching (Ainsworth et al., 2023) to a partial matching, followed by a layer-wise least-squares refit of the merged weights. Partial fusion differs in that the matching is

based on (partial) optimal transport, which contains partial permutations as a special case and extends to layers of unequal size, and in that the merged weights arise purely from weight interpolation, without a data-dependent refitting step. Moreover, our generalized pruning perspective (Section 2.3) places both approaches in a common framework alongside pruning and clustering of ensembles.

The direct partial fusion approach described above can most easily be interpreted as considering normal model fusion as the baseline, and then relaxing the method by fusing fewer neurons. We additionally consider the opposing point of view: Instead of taking model fusion as the baseline and relaxing it, we take the ensemble model as the baseline and then apply a certain pruning operation. We call these operations *generalized pruning*, which are similar to layer-wise pruning, but instead of merely deleting neurons we can also linearly combine them. The way we formalize generalized pruning of ensembles in fact leads to more degrees of freedom than the partial fusion method described above. Indeed, partial fusion can be interpreted as a particular inductive bias when applying generalized pruning to the ensemble, see Section 2.3. Beyond this particular case, we showcase one further particular method to realize generalized pruning via clustering, see Section 2.4. Compared to just pruning (or pruning and post-processing), the flexibility of generalized pruning via clustering allows compression of models with smaller drops in performance, see Figure 1b.

In terms of broader understanding of neural networks, we believe both viewpoints described above, relaxing weight aggregation and generalized pruning, are building on the following observation: Keeping only the least similar neurons isolated increases the average similarity of those neurons which are aggregated (cf. Appendix L). Hence, the primary insight of our results in this regard is that, first of all, the difference in similarities between neurons (i.e., how unique or non-unique the role of a neuron is) is significant, both across different networks (for model aggregation) and within single networks (for generalized pruning). And second, the differences in similarity are not just artifacts of how we measure similarity, but can often efficiently be exploited by the proposed partial fusion and generalized pruning methods to navigate the tradeoff between model size and performance. To summarize, the main contributions of the paper are:

- In Section 2.2, we introduce *partial fusion* to interpolate between weight aggregation and ensembles by only merging the most similar neurons. This provides a flexible tradeoff between model size and performance.

- In Section 2.3, we show that partial fusion is a special case of *generalized pruning* of ensembles, where neurons may be linearly combined rather than just deleted. This perspective extends to single-model compression.

- In Section 2.4, we give concrete methods to perform partial fusion via partial optimal transport and generalized pruning via clustering. The former extends prior work on OT-based model fusion (Singh & Jaggi, 2020).

- In Section 3, experiments on MLPs and CNNs show that the tradeoffs are often efficient in the sense that an increase in model size leads to disproportionate performance gains when going from weight aggregation to ensembles. For single-model pruning, generalized pruning via clustering outperforms standard pruning and pruning with post-processing.

## 2. Partial fusion and generalized pruning of networks

In this section, let $A$ and $B$ be $L$-Layer feedforward NNs giving rise to functions $f_A, f_B : \mathbb{R}^{n_0} \to \mathbb{R}^{n_{L+1}}$, with hidden dimensions $n_\ell^A, n_\ell^B$ and weights $W_\ell^A \in \mathbb{R}^{n_{\ell+1}^A \times n_\ell^A}, W_\ell^B \in \mathbb{R}^{n_{\ell+1}^B \times n_\ell^B}$ (for $\ell = 0, \dots, L$), with $n_0^A = n_0^B = n_0$ and $n_{L+1}^A = n_{L+1}^B = n_{L+1}$, without bias terms. For more general architectures, we refer to Appendix E.

### 2.1. Background on model fusion

In model fusion (or weight aggregation), the goal is to define a new neural network $C$ with weights that are aggregates of those of $A$ and $B$.

In the line of research for layer-by-layer model fusion which we follow, for each layer $\ell$ we require a way to translate the state $\mathbb{R}^{n_\ell^A}$ of hidden neurons of Model $A$ into the state $\mathbb{R}^{n_\ell^B}$ of Model $B$, and vice versa. This is modeled by linear transformations, or matrices, $K_\ell^{A \to B} \in \mathbb{R}^{n_\ell^B \times n_\ell^A}$ and $K_\ell^{B \to A} \in \mathbb{R}^{n_\ell^A \times n_\ell^B}$. These matrices can intuitively be regarded as basis transformations, though particularly with unequal dimensions, more general transformations are used.

Given such (basis) transformations, model fusion becomes simple: While the weight matrix $W_\ell^A$ corresponds to a linear transformation in the world of network $A$, it can be translated into the world of $B$ via

$$\widetilde{W}_\ell^A := K_{\ell+1}^{A \to B} W_\ell^A K_\ell^{B \to A},$$

see Figure 3.

Fusing the $\ell$-th layer weights of both models $A$ and $B$ into the world of $B$ is thus simply done by

$$W_\ell^C = (1 - \lambda) W_\ell^B + \lambda \widetilde{W}_\ell^A, \tag{1}$$

where $\lambda \in [0, 1]$ is the interpolation factor which assigns weights to the initial networks, see Appendix K regarding a discussion of the parameter $\lambda$. The network $C$ which is defined through the weight matrices $W_\ell^C$ thus is a fused

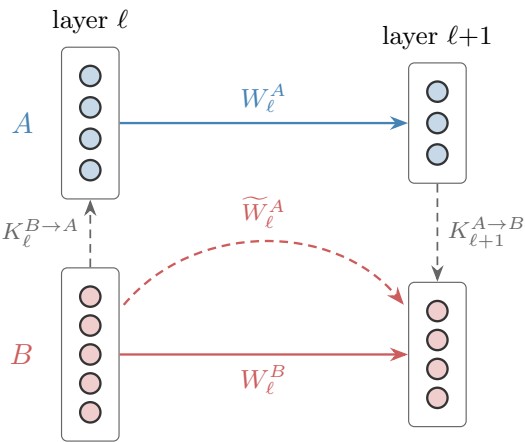

layer $\ell$     layer $\ell+1$

*Figure 3.* The weight matrix $W_\ell^A$ of network $A$ induces a weight matrix $\widetilde{W}_\ell^A = K_{\ell+1}^{A\to B} W_\ell^A K_\ell^{B\to A}$ between the neurons of network $B$ via the transformations $K_\ell^{B\to A}$ and $K_{\ell+1}^{A\to B}$.

version of $A$ and $B$, while (by convention) seen as belonging to the world of Model $B$. In particular, the layer sizes of model $C$ will be precisely the same as that of Model $B$.

Existing methods focus for instance on permutation matrices $K_\ell^{A\to B} = P_\ell$ to align neurons between the different networks, in which case $K_\ell^{B\to A} = P_\ell^T$ is usually simply the inverse transformation (see, e.g., Ainsworth et al. (2023); Jordan et al. (2023); Stoica et al. (2024); Entezari et al. (2022)). While permutation matrices only work for equally sized layers, a natural generalization is via column-stochastic matrices, which for instance arise via stochastic kernels and optimal transport (see, e.g., Akash et al. (2022); Ferbach et al. (2024); Imfeld et al. (2024); Singh & Jaggi (2020)).

## 2.2. Partial fusion of neural networks

The main idea of partial fusion is to only combine *parts* of the weights of each layer. This is illustrated in Figure 2, which showcases an approach based on permutation alignment between neurons of two shallow networks.

To formalize this, in each layer $\ell$ we are given indices $I_\ell^A \subseteq \{1,\ldots,n_\ell^A\}, I_\ell^B \subseteq \{1,\ldots,n_\ell^B\}$ of neurons which are left isolated, and correspondingly the sets of neurons which will be fused, which are simply the remaining neurons $F_\ell^A = \{1,\ldots,n_\ell^A\} \setminus I_\ell^A, F_\ell^B = \{1,\ldots,n_\ell^B\} \setminus I_\ell^B$. When clear from context, we abbreviate $I = I_\ell^A$ etc., for instance in $W_\ell^A[F,I] := W_\ell^A[F_{\ell+1}^A, I_\ell^A]$. The transformations between networks are then only between the fused neurons, so for instance $K_\ell^{A\to B} \in \mathbb{R}^{F_\ell^B \times F_\ell^A}$, and correspondingly

$$\widetilde{W}_\ell^A := K_{\ell+1}^{A\to B} W_\ell^A[F,F] K_\ell^{B\to A}$$

only describes the weights between fused neurons.

The partially fused network, which we will again call $C$, keeps the isolated parts of both initial networks, while fusing only those neurons which belong to the fused parts. As in normal model fusion, the fused neurons are (by convention) considered as belonging to the world of Model $B$, and thus the $\ell$-th layer of the fused network lives in three different parts: First, the isolated part of Model $A$, so $I_\ell^A$. Second, the fused part, which is within the world of $B$, so $F_\ell^B$. And third, the isolated part of $B$, so $I_\ell^B$. We can define the overall matrix $W_\ell^C$ of the partially fused network $C$ from layer $\ell$ to $\ell+1$ as shown in Figure 4.

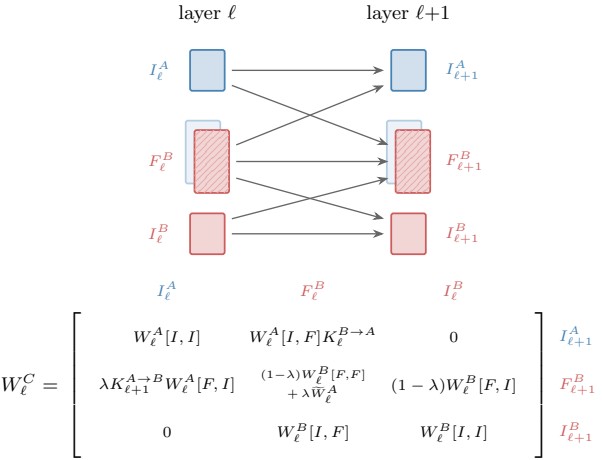

layer $\ell$     layer $\ell+1$

$$W_\ell^C = \begin{bmatrix} W_\ell^A[I,I] & W_\ell^A[I,F]K^{B\to A} & 0 \\ \lambda K_{\ell+1}^{A\to B} W_\ell^A[F,I] & \begin{matrix}(1-\lambda)W_\ell^B[F,F]\\ + \lambda \widetilde{W}_\ell^A\end{matrix} & (1-\lambda)W_\ell^B[F,I] \\ 0 & W_\ell^B[I,F] & W_\ell^B[I,I] \end{bmatrix} \begin{matrix} I_{\ell+1}^A \\ F_{\ell+1}^B \\ I_{\ell+1}^B \end{matrix}$$

*Figure 4.* Visualization of a partially fused layer and definition of the corresponding weight matrix.

We emphasize that each of the seven arrows in the diagram of Figure 4 corresponds to one non-zero block in the weight matrix $W_\ell^C$ of the fused network. The diagonal blocks are easily interpreted: The top left and bottom right are simply isolated parts of the initial models, and in the middle we have the same combination of weights as in normal model fusion, cf. equation (1). The off-diagonal blocks are the maps between isolated and fused parts, where the transition in and out of the isolated parts of $A$ again require the transformations $K_\ell^{B\to A}$ and $K_{\ell+1}^{A\to B}$. In terms of the interpolation factor $\lambda \in [0,1]$, the logic is the same as in ensembles: When going *into* the fused part, the weighting must be applied.

## 2.3. Generalized pruning of networks

Abstractly, partial fusion can be regarded as a flexible way to embed the concatenated layers of $A$ and $B$ into a joint, smaller, network. This is not just reminiscent of pruning networks, but below we also show that it is formally a special case, at least if pruning is interpreted in a suitably general way.

To this end, let $E$ be an $L$-layer NN (with the same notational conventions as for $A$ and $B$). The model $E$ should

be interpreted as a large model, for instance an ensemble. In what we call generalized pruning, we are given transformations between the world of $E$ and the world of an implicitly defined, smaller network $S$, that is, we are given $K_\ell^{E \to S} \in \mathbb{R}^{n_\ell^S \times n_\ell^E}$ and $K_\ell^{S \to E} \in \mathbb{R}^{n_\ell^E \times n_\ell^S}$. As in standard model fusion (cf. Figure 3), we transfer the weights of $E$ into the world of $S$ via

$$W_\ell^S := K_{\ell+1}^{E \to S} W_\ell^E K_\ell^{S \to E}.$$

This construction indeed contains standard pruning of neural networks as a special case if $K_{\ell+1}^{E \to S} \in \{0,1\}^{n_{\ell+1}^S \times n_{\ell+1}^E}$ is a row-stochastic matrix which deletes certain rows of $W_\ell^E$, and $K_\ell^{S \to E} \in \{0,1\}^{n_\ell^E \times n_\ell^S}$ is a column-stochastic matrix which deletes certain columns. More general choices of such transformations, in particular stochastic matrices related to clustering, are discussed below in Section 2.4.

Partial fusion of two models $A$ and $B$ arises as a particular generalized pruning applied to the ensemble of the networks $A$ and $B$. While the precise form of the matrices $K_\ell^{E \to S}$ and $K_\ell^{S \to E}$ is given in Appendix G, we shortly discuss the specific inductive bias given by partial fusion within the space of all possible generalized pruning methods.

Compared to arbitrary generalized pruning, partial fusion only ever linearly combines neurons if they initially belonged to different networks, and at most two neurons are linearly combined instead of arbitrarily many. This inductive bias should be valuable if neurons (or channels) from different networks are more likely to have similar functional roles than neurons within the same network (see Appendix L for an exploration of this hypothetical). The clustering method for generalized pruning introduced in the following subsection lacks this bias. The relative performance of these methods observed in the experiments (see Figures 10 and 11 in the Appendix) suggest that this inductive bias of partial fusion could be more useful for CNNs compared to MLPs.

We shortly emphasize that Stoica et al. (2024) consider a similarly flexible viewpoint to model merging as generalized pruning, in that they first concatenate layers and then apply a flexible merge operation to the concatenation.

### 2.4. Transformations between different layers

To define transformations between layers, we follow Singh & Jaggi (2020) by encoding a layer as a discrete probability distribution, where the possible states are the features of the neurons. So a layer with $n$ neurons is encoded by a discrete probability measure supported on $n$ points. The support (or state space) is governed by the choice of the feature vectors of the neurons, which are usually defined through activation vectors or in- and outgoing weights of a neuron.

So for two networks $A$ and $B$ and a layer $\ell$, we are given feature vectors $x_\ell^A \in (\mathbb{R}^{d_\ell})^{n_\ell^A}$, $x_\ell^B \in (\mathbb{R}^{d_\ell})^{n_\ell^B}$ with corre-

sponding probabilities $\mu_\ell^A \in [0,1]^{n_\ell^A}$ and $\mu_\ell^B \in [0,1]^{n_\ell^B}$. Hereby, one usually simply assigns uniform weights across neurons, but in general the assigned weight can be derived, e.g., from a notion of importance of a neuron. A coupling $\pi$ between $\mu_\ell^A$ and $\mu_\ell^B$ is a matrix $\pi \in [0,1]^{n_\ell^A \times n_\ell^B}$ satisfying $\sum_{j=1}^{n_\ell^B} \pi[i,j] = \mu_\ell^A[i]$ and $\sum_{i=1}^{n_\ell^A} \pi[i,j] = \mu_\ell^B[j]$; we denote the set of such couplings by $\Pi(\mu_\ell^A, \mu_\ell^B)$. Each such coupling $\pi_\ell^{A,B}$ implicitly defines two transformations (or stochastic kernels) given by, with NumPy notation,

$$\begin{aligned} K_\ell^{A \to B} &= (\pi_\ell^{A,B})^T / \mu_\ell^A[\text{None}, :], \\ K_\ell^{B \to A} &= \pi_\ell^{A,B} / \mu_\ell^B[\text{None}, :]. \end{aligned} \tag{2}$$

With these notations in place, model fusion as in (Singh & Jaggi, 2020) arises via the optimal transport problem

$$\pi_\ell^{A,B} = \operatorname*{argmin}_{\pi_\ell \in \Pi(\mu_\ell^A, \mu_\ell^B)} \int \|x - y\|^2 \, \pi_\ell(dx, dy)$$

and we can obtain transformations for generalized pruning by the clustering objective

$$\pi_\ell^{E,S} = \operatorname*{argmin}_{\pi_\ell \in \Pi(\mu_\ell^E, *_m)} \int \|x - y\|^2 \, \pi_\ell(dx, dy),$$

where $\Pi(\mu_\ell^E, *_m)$ denotes the set of couplings with unspecified second marginal supported on at most $m$ points ($m$ is the desired number of cluster centers). For a uniform distribution $\mu_\ell^E$, this is the standard K-means clustering objective. The optimizer $\pi_\ell^{E,S}$, resp. the corresponding disintegration $K_\ell^{E \to S}$, is simply given by the map which assigns points to their respective cluster centers. We emphasize that in the regimes where this objective is relevant for generalized pruning, the standard K-means algorithm (Lloyd's algorithm) performs badly and hierarchical clustering algorithms (see Appendix J) yield solutions closer to the global optimum. Further, we mention that intriguingly, Luenam et al. (2025) recently introduced a similar clustering objective in the context of model aggregation, though their goal was not to use it for pruning, but to aggregate weights from many networks.

To extend OT Fusion (Singh & Jaggi, 2020) to *Partial OT Fusion* with parameter $\alpha \in [0,1]$ we allow for couplings which only match $(1-\alpha)$ of the mass between two marginals, cf. Figalli (2010). This means partially fusing two equally sized layers leads to $(1+\alpha)\times$ the neurons in the fused layer. An $\alpha$-partial coupling $\tilde{\pi} \in [0,1]^{n_\ell^A \times n_\ell^B}$ satisfies $\sum_{j=1}^{n_\ell^B} \tilde{\pi}[i,j] \le \mu_\ell^A[i], \sum_{i=1}^{n_\ell^A} \tilde{\pi}[i,j] \le \mu_\ell^B[j]$ and all entries of $\tilde{\pi}$ sum to $(1-\alpha)$; we denote the corresponding set of $\alpha$-partial couplings by $\Pi_\alpha(\mu_\ell^A, \mu_\ell^B)$. The way we obtain isolated neurons and transformations for the partial model fusion is through the partial optimal transport problem

$$\tilde{\pi}_\ell^{A,B} := \operatorname*{argmin}_{\pi_\ell \in \Pi_\alpha(\mu_\ell^A, \mu_\ell^B)} \int \|x - y\|^2 \, \pi_\ell(dx, dy). \tag{3}$$

This problem is no more difficult to solve than a normal optimal transport problem, and in fact equivalent to one. Having solved the problem, the isolated neurons of network $A$ and $B$ are then simply those neurons which, according to $\tilde{\pi}_\ell^{A,B}$, are not matched (how to deal with partially matched neurons is explained in Appendix A). The transformation between non-isolated neurons is obtained as in (2) by defining $\pi_\ell^{A,B}$ as the restriction to non-isolated neurons of $\tilde{\pi}_\ell^{A,B}$, normalizing it to one, and suitably dividing it by its own marginals to obtain the stochastic kernels (similar to (2)).

### 2.5. Beyond layer-by-layer matchings

An insight from Ainsworth et al. (2023) is that when accounting for permutation invariance of different networks' neurons, one should optimize these permutations globally across layers, and not just for each layer in isolation. In the terminology of this paper, this means that allowing the features of neurons in one layer to depend on the transformations between neurons in the other layers can lead to significantly improved fusion of networks in total. The drawback is that to obtain optimal transformations then requires a joint optimization across all layers. To formalize this, the feature vectors $x_\ell^A$ and the resulting probabilities $\mu_\ell^A$ depend on the (partial) matchings of other layers. That is, with a slight abuse of notation,

$$x_\ell^A = x_\ell^A((\pi_{\tilde{\ell}}^{A,B})_{\tilde{\ell} \neq \ell}) \quad \text{and} \quad \mu_\ell^A = \mu_\ell^A((\pi_{\tilde{\ell}}^{A,B})_{\tilde{\ell} \neq \ell}).$$

For instance, in normal fusion, one such dependence arises from using weight matrices as features. In this case, to use $W_\ell^A$ and $W_\ell^B$ as a feature in layer $\ell$ already requires an alignment of the output space (i.e., of layer $\ell + 1$) of the matrices, leading to

$$x_\ell^A = x_\ell^A(\pi_{\ell+1}^{A,B}) = K_{\ell+1}^{A \to B} W_\ell^A.$$

In general, including this type of dependence for normal fusion leads to the global objective

$$\left(\pi_\ell^{A,B}\right)_{\ell=1}^L = \operatorname*{argmin}_{\substack{\pi_\ell \in \Pi(\mu_\ell^A, \mu_\ell^B), \\ \ell=1,\ldots,L}} \sum_{\ell=1}^L \int \|x - y\|^2 \, \pi_\ell(dx, dy).$$

To approximately solve this, one can either follow a greedy approach (as used in Singh & Jaggi (2020)), or a fixed-point iteration (as introduced by Ainsworth et al. (2023) in case of permutations). In each step, the fixed-point iteration updates one particular $\pi_\ell$ while keeping all others unchanged. By restricting the optimization in each iteration to the terms depending linearly on $\pi_\ell$, this again leads to an optimal transport problem in each step and can thus be solved efficiently. For the most relevant case, in which features arise from weight matrices, the linear objective in each fixed-point iteration captures most of the global objective, see Appendix D.2.

Extending this approach to partial fusion is straightforward with the constraints and objective as in (3) given by partial optimal transport, see again Appendix D.2 in case that weight matrices are used to define the features.

## 3. Experiments

We showcase the introduced methods on commonly used toy examples. In Section 3.1, we compare different methods for partially aggregating two MLPs which were both trained from scratch on different parts of MNIST, which is a regime where model aggregation is plausibly very useful. In Section 3.2 we examine how the performance of aggregated models can be further improved by fine-tuning the model with a small sample of the overall data distribution. We evaluate the fine-tuning of the partially fused MLP models, as well as the fine-tuning of two partially fused ResNet18 models, trained on heterogeneous data-splits of CIFAR10. In Subsection 3.3, we treat aggregation of models (both MLPs on MNIST and CNNs on CIFAR10) which were trained on the same data with different random seeds. In Section 3.4, we compare different methods in the classic unstructured pruning setting to illustrate the efficacy of the introduced generalized pruning for just a single model.

For implementation details, we refer to Appendix A.

### 3.1. Model Aggregation: Split Dataset

We consider a setting introduced by Singh & Jaggi (2020), in which we have a heterogeneous data-split of MNIST. The first split contains all of the samples for one specific digit (e.g., the digit 4) and 10% of the samples of all other digits. The other split contains the remaining 90% of the samples that are not the specific digit. Model A is trained on the first split and model B on the second split. The goal is to simulate a setting in which we have a general model (Model B), which is good on most of the data and a specialized model (Model A) that has knowledge the general model does not possess. In this setting fusion of the models can lead to knowledge transfer, resulting in a fused model with better performance than both individual models. For our experiments we trained five pairs of MLP models and all reported values are averages over the five resulting aggregated models.

Results for different matching strategies for Partial OT Fusion are shown in Figure 5. The values for $\alpha$ used are $0, 0.2, 0.4, 0.5, 0.6, 0.8, 1$ in each plot, which give relative increase in neuron count compared to single models ($\alpha = 0$ yields the size of a single model, and $\alpha = 1$ that of the ensemble). We emphasize that the increase in neurons is not proportional to increase in effective parameters, see Appendix H. In Figure 5, the Partial OT Fusion method using the fixed-point approach (cf. Appendix D) outperforms the

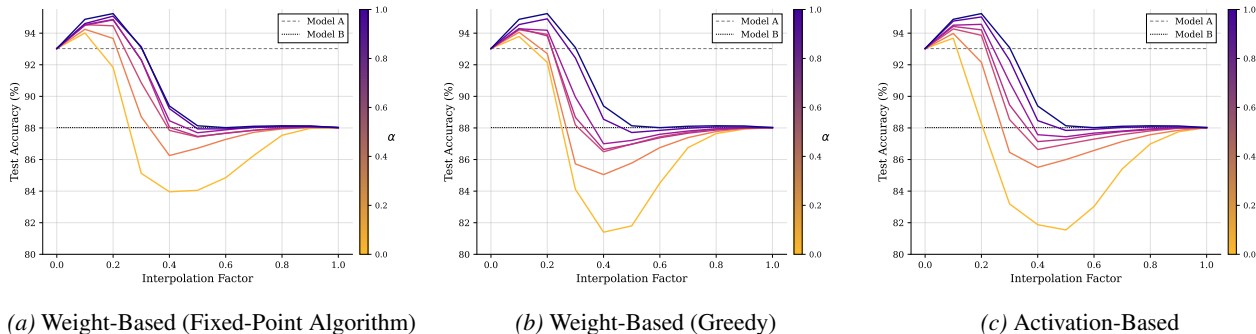

*(a)* Weight-Based (Fixed-Point Algorithm)     *(b)* Weight-Based (Greedy)     *(c)* Activation-Based

*Figure 5.* Partial OT Fusion of two MLPs $A$ and $B$ trained on different parts of MNIST. The Interpolation Factor determines the weight given to $A$ and $B$. The factor $\alpha$ determines the number of neurons in the fused model ($\alpha = 0$ is weight aggregation and $\alpha = 1$ is the ensemble). Panels (a), (b) and (c) arise from different specifications of the partial OT problem. Panels (a) and (b) use weight matrices as features (cf. Section 2.5 and Appendix D), but (a) uses the fixed-point algorithm while (b) uses greedy matching; Panel (c) uses activation vectors. Weight-based fusion using the fixed-point algorithm yields the highest accuracies. All panels show that leaving only a small fraction of neurons isolated often yields a large increase in accuracy. Reported values are averages over five random seeds.

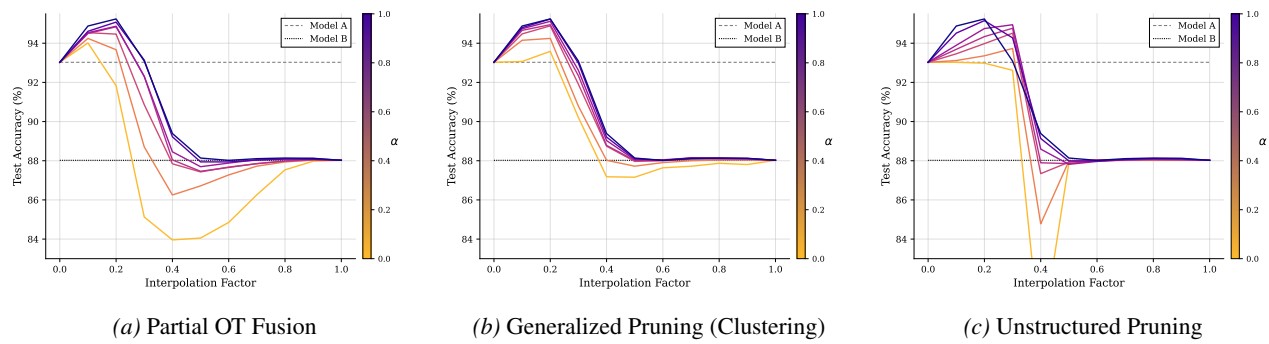

*(a)* Partial OT Fusion     *(b)* Generalized Pruning (Clustering)     *(c)* Unstructured Pruning

*Figure 6.* Model aggregation of two MLP models (as in Figure 5) using either Partial OT Fusion (Panel (a); as in Figure 5a), generalized pruning (based on clustering, Panel (b)) or unstructured pruning (Panel (c)) of the ensemble. Generalized Pruning often outperforms Partial OT Fusion, but the clustering objective is more costly to solve. Unstructured pruning shows no clear trend compared with the other methods, but the results for certain interpolation factors seem promising as well, making it a simple cost-efficient alternative to clustering.

greedy weight-based and activation-based methods. This shows that incorporating the fixed-point iteration introduced by Ainsworth et al. (2023) into the OT Fusion method of Singh & Jaggi (2020) is beneficial.

Figure 6 shows a comparison between Partial OT Fusion (using the fixed-point algorithm as in Figure 5, Panel (a)), and (generalized) pruning of the ensemble.

Partial OT Fusion provides an efficient interpolation between OT fusion and ensembles. A $20\%$ or $40\%$ increase in neurons per layer already yields a clear improvement over standard OT fusion (the $40\%$ case is the one reported in Figure 1a which leads to roughly a $45\%$ increase in parameters). Throughout, performance improves monotonically with the number of neurons, indicating that Partial OT Fusion is a natural extension of OT Fusion.

Generalized pruning via clustering often improves the trade-off between parameters and performance over Partial OT Fusion, particularly for larger values of $\alpha$ (near the ensemble). This increase in performance for larger $\alpha$ comes at the

cost of approximately solving the NP-hard problem of clustering (Aloise et al., 2009) to aggregate neurons. Further, when restricted to the size of the original models ($\alpha = 0$), the models arising from generalized pruning are close to the initial models and show less variability compared to the OT Fusion model, importantly also in the case near Model $A$.

Unstructured pruning of the ensemble does not yield a monotonic interpolation between a single pruned model and the full ensemble. Particularly at an interpolation factor of $0.3$, unstructured pruning surpasses the full ensemble, indicating non-monotonic behavior. A possible explanation is that our current method for pruning an ensemble overemphasizes the interpolation factor to be more extreme, as it enters once through the weighting in the output layer and once through the assigned importance of neurons, see Appendix F. Thus, an interpolation factor of $0.3$ for partial fusion may be more similar to the interpolation factor of $0.1$ or $0.2$ for the ensemble model, explaining the improved performance.

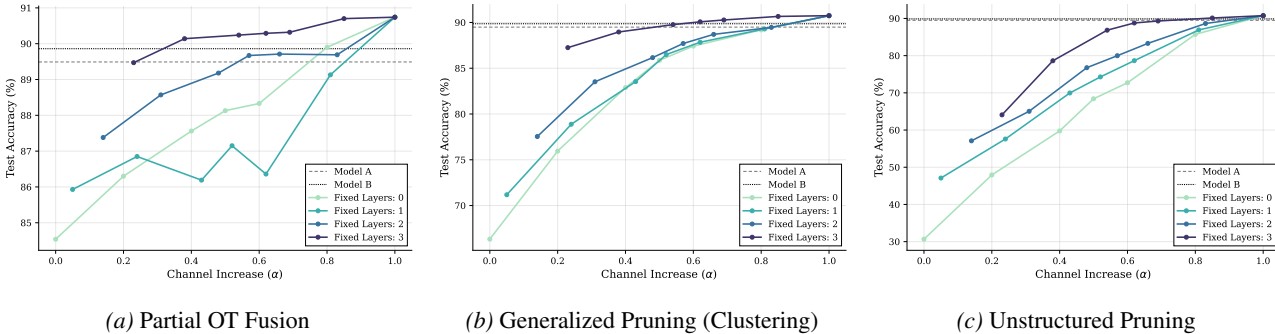

| (a) Partial OT Fusion | (b) Generalized Pruning (Clustering) | (c) Unstructured Pruning |

*Figure 7.* Equally weighted model aggregation of two CNN models with the methods as in Figure 6, while keeping a variable amount of layers fixed (at the size of the ensemble). The $x$-axis gives the number of channels *including* the frozen layers. Keeping some layers fixed leads to better accuracies for a fixed amount of channels, indicating that different layers have different suitability for model aggregation.

### 3.2. Model Aggregation and Fine-tuning

In this section we consider again a setting in which two models are trained on heterogeneous data-splits, but now a small subsample of the overall data distribution is available, that can be used to re-train the aggregated model. We consider the same two MLP models trained on MNIST as in Section 3.1 and use 1% of the training data of all MNIST classes to re-train the fused model. Additionally we evaluate the partial fusion of two ResNet18 models (He et al., 2016) trained on heterogeneous data splits of CIFAR10. 5% of all the data is held out, for fine-tuning of fused model. Model A is trained on 92% of the data of all classes with an even index and 3% of the data of all other classes. Model B is trained on the remaining data. We note that when fine-tuning partially fused models all zero block-matrices introduced by partial fusion (see Figure 4) are frozen and thus the parameter count and the count of non-zero parameters does not increase during fine-tuning.

Table 1 shows the results after fine-tuning the partially fused models. For both the MLP and ResNet models, the fine-tuned partially fused models surpasses the individual models trained on the heterogeneous data-splits for all $\alpha$. Increasing the number of isolated neurons increases the performance of the fused model, both before and after fine-tuning. For more refined baselines, which include fine-tuning the individual models, see Appendix B.2.1.

### 3.3. Model Aggregation: Same Data

We consider the setting of two networks that are independently trained on the same dataset. In this setting, model aggregation overall leads to fewer benefits compared to Subsection 3.1. The baseline results by partial aggregation for two fused VGG11 models on CIFAR10 and two fused MLPs on MNIST are shown in Appendix B.

Figure 8 shows the partial fusion of two ResNet18 models trained on CIFAR10. Again partial fusion interpolates be-

*Table 1.* Accuracy (%) after fine-tuning partially fused models under limited data. The first block uses MLPs trained on split MNIST datasets as in Section 3.1; the second uses two ResNet-18 models trained on a similar split CIFAR-10 dataset. Models are fused with $\lambda = 0.5$ and fine-tuned with 1% of MNIST or 5% of CIFAR-10 training data. Fine-tuning the partially fused models consistently surpasses the individual models. For more refined baselines see Appendix B.2.1 (in particular Table 3).

| $\alpha =$ | 0.0 | 0.2 | 0.4 | 0.5 | 0.6 | 0.8 | 1.0 |
|---|---|---|---|---|---|---|---|
| **MLP on MNIST (1% of data)** | | | | | | | |
| Fusion | 84.1 | 86.7 | 87.4 | 87.5 | 87.7 | 87.9 | 88.1 |
| Fine-tuning | 95.1 | 95.7 | 96.1 | 96.2 | 96.3 | 96.5 | 96.5 |
| *Reference: Model A/B: 93.8/87.8* | | | | | | | |
| **ResNet-18 on CIFAR-10 (5% of data)** | | | | | | | |
| Fusion | 66.4 | 79.2 | 83.4 | 87.4 | 88.9 | 90.6 | 91.3 |
| Fine-tuning | 85.3 | 88.3 | 90.0 | 90.3 | 90.8 | 91.4 | 91.8 |
| *Reference: Model A/B: 79.8/76.7* | | | | | | | |

tween OT fusion and ensembles. Compared to the fusion of VGG11 models, the completely fused ResNet18 models are much closer in performance to the individual models. This is likely due to the tied permutation invariance structure of layers that are connected by residual connections, which we exploit when computing fusion kernels for ResNet models (see Appendix E.1). However we also note that the performance of the partially fused models depends on how the individual models are trained (see Appendix B.1).

To study the possibilities of the introduced methods even in this setting, we consider separate treatment of different layers of VGG11 models. Since fusion of layers is generally easier for wider layers (Ainsworth et al., 2023), this motivates the use of different $\alpha$ depending on the layer width. As a first step we simply explore this possibility by keeping one (layer 2), two (layer 2 and 3) or three (layer 2, 3 and 4) of the eight convolutional layers fixed as an ensemble (so at $\alpha = 1$), while partially aggregating the rest of the layers.

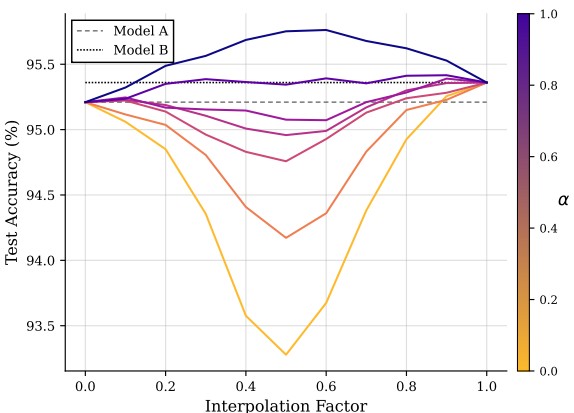

*Figure 8.* Partial fusion of two ResNet18 models trained independently on CIFAR10. The decrease in performance when fusing the models compared to the individual models is much smaller compared to the fusion of VGG11 models (see Figure 11a). Reported values are averaged over five random seeds.

Results for an interpolation factor $0.5$ are shown in Figure 7. We see that fixing layers can improve the performance of all methods. Fixing one layer leads to mixed results, with improvements for unstructured pruning but unclear tendencies for partial fusion and clustering. However, fixing three layers leads to significant improvements for all methods. Partial fusion exceeds the performance of both individual models with an increase of the overall channel count by only $38\%$.

### 3.4. Generalized Pruning of a Single Model

In this section we consider a single neural network, with the goal of reducing the number of neurons in each layer. We compare a general clustering approach to unstructured pruning and unstructured pruning combined with fusion based post-processing introduced by Singh & Jaggi (2020). Results for an MLP on MNIST are reported in Figure 1b and for CNNs on CIFAR10 in Figure 9. Both OT Fusion post-processing and clustering improve over simple unstructured pruning for VGG11. Clustering slightly improves over OT Fusion post-processing, but again comes at the cost of increased computational costs to (approximately) solve the clustering problem.

For pruning an MLP on MNIST, shown in Figure 1b, clustering shows significant improvements compared to unstructured pruning and OT Fusion. One possible explanation is that we can distinguish between two different ways leading to a decrease in accuracy when pruning in the general sense: Either a neuron is deleted (processing steps are lost), or two neurons are linearly combined (processing steps are blurred). In particular, pruning only has the first type of error (deletion of neurons), while pruning together with post-processing by OT Fusion only has the second type of error (blurring). In

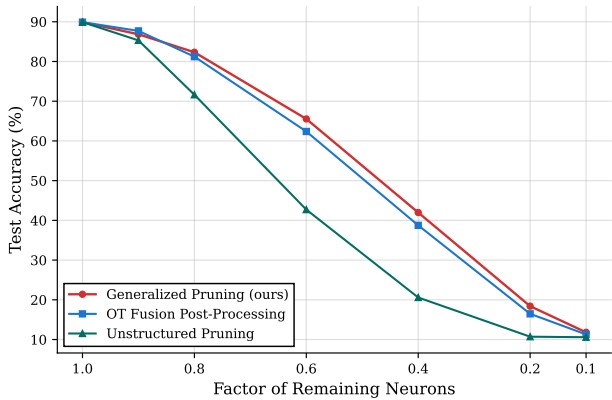

*Figure 9.* Comparison of unstructured pruning (with or without post-processing) with generalized pruning (based on clustering) for a single VGG11 model trained on CIFAR10.

Figure 1b, there are presumably two regimes, one (factor $> 0.4$) where blurring is more costly, and one (factor $< 0.4$) where deletion is more costly. Clustering leads to a more flexible tradeoff between these two sources of errors, and thus outperforms the other methods. This interpretation is consistent with Figure 9, since therein blurring appears less costly throughout, and thus clustering does not yield such a large benefit compared to post-processing.

## 4. Discussion and Future Work

This paper shows that different tradeoffs between weight aggregation and ensembles are possible, by either extending weight aggregation methods (e.g., via Partial OT Fusion), or by generalized pruning of an ensemble (e.g., via clustering). The experiments showcase the feasibility and often the efficiency of the resulting tradeoffs. For future work, we see two interesting research directions: First, partial OT Fusion and clustering only exploit permutation invariance and perhaps richer notions of similarity between neurons can be exploited to improve performance, for instance based on CCA (Horoi et al., 2024; Kornblith et al., 2019). Second, our experiments are limited to standard benchmarks with small architectures; scaling to larger models in combination with fine-tuning and more fine-grained use of the methods remains to be explored. In this regard, Figure 7 suggests that layer-dependent treatment can improve performance, and developing principled criteria for such layer-wise decisions is a promising direction. Understanding which layers are most amenable to fusion may also yield insights into the functional roles of neurons across different layers.

## Acknowledgments

We thank the reviewers for their thorough reading and diverse perspectives, which in particular helped us improve the empirical support for our proposed methods. The authors are grateful for support by the German Research Foundation through Project 553088969 as well as the Cluster of Excellence "Machine Learning — New Perspectives for Science" (EXC 2064/1 number 390727645).

## Impact Statement

This paper presents work whose goal is to advance the field of Machine Learning. There are many potential societal consequences of our work, none which we feel must be specifically highlighted here.

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

# A. Experimental details

Unless otherwise stated, Partial OT Fusion uses the weights of the layers as features and is optimized with the projected coordinate descent (or fixed-point) approach described in Section 2.5 and Appendix D.2. For Partial OT Fusion, this choice was tuned as in Figure 5; in contrast, we always use activation vectors as features for clustering. While this choice could be tuned as well, the main comparison (cf. Figure 6) favors the clustering already with activation vectors, so we leave a more detailed comparison of different features for clustering to future work. For the clustering algorithm we use stochastic hierarchical clustering, with Ward's method (Ward Jr., 1963) as the criterion, see Appendix J. For the unstructured pruning we use the $L_2$ norm of the weights as the pruning criterion (Cheng et al., 2024; He & Xiao, 2024).

The MLP models on MNIST (LeCun et al., 1998) have three hidden layers with 100 neurons each. The models use the GELU activation function to mitigate the effect of dead neurons when pruning. The models are trained using the Adam optimizer with a fixed learning rate of 0.001 for 50 epochs.

For CNNs, we use standard VGG11 networks (as used in Singh & Jaggi (2020), see also (Simonyan & Zisserman, 2015)) trained with SGD for 300 epochs. The initial learning rate is set to 0.5, with a decay by factor 2 every 30 epochs. The momentum is set to 0.9 and weight decay to 0.0005. We use early stopping to choose the best checkpoint.

When using activation vectors as features for neurons in either partial OT Fusion or clustering, this is done similarly as in (Singh & Jaggi, 2020) using 1000 data points from the training set. We varied the number of data points and found that mainly, a higher number slightly reduces noise across different seeds, but otherwise makes little difference. Further, whether data from training or test set was used showed no clear differences. Notably, we always use activation vectors for generalized pruning via clustering, as this seemed to be the most natural choice of features for this task.

Finally, we mention that partial optimal transport a priori does not automatically lead to hard partitions between isolated and fused neurons for partial fusion (cf. Section 2.4). In practice, we observe that always at most a single neuron belongs to both isolated and fused part according to the partial optimal transport coupling. Hence, while the way this case is treated is not of large practical importance, we detail our procedure nevertheless in the following: A partially matched neuron is treated as if there were actually two neurons which, together, have the same role as the one neuron which was partially matched. Then one of these two neurons is put into the fused part and one into the isolated part. That is, if a neuron $i$ of net $A$ is given total weight

$\kappa < \mu_\ell^A[i]$ by the partial optimal transport coupling, we split it into two new neurons. These new neurons have as incoming weights the same incoming weights as the initial neuron, but scaled by the factors $\frac{\kappa}{\mu_\ell^A[i]}$ and $\frac{\mu_\ell^A[i]-\kappa}{\mu_\ell^A[i]}$. The output weights for the new neurons are the same as for the initial neuron. We emphasize that, for a RELU network, due to positive-homogeneity, such a split always leads to the same values propagated to the next layer. While this is not exactly true for GELU, we think it is still a good approximation.

# B. Further results

In this section we show additional results for generalized pruning.

Figure 10 shows the results for averaging pairs of MLP models trained on MNIST. All three methods benefit from increasing the number of neurons in each layer. As with other experiments with MLPs, clustering outperforms the other methods. The figures also suggest that weight aggregation via OT Fusion, and consequently also partial OT Fusion, is more directly practically useful in a setting as in Subsection 3.1 when models to be merged were trained with heterogeneous data.

Figure 11 shows the results for averaging pairs of VGG11 models trained on CIFAR10. Partial OT Fusion outperforms clustering and unstructured pruning on the task of aggregating two VGG11 models. Partial OT Fusion shows the strongest performance of all methods when completely fusing the networks ($\alpha = 0$), but it also benefits from an increase in neurons, improving its accuracy by 1.8 percentage points, with an increase of 20% in neuron count. Clustering and especially pruning fail to preserve the performance of the models, when fully fusing the two models with equal importance, dropping to 66.3% and 30.7% accuracy respectively. Both methods increase the performance with increasing neurons, but do not match the performance of partial fusion. This indicates that the restriction of partial fusion to always match two individual channels might be an important bias for fusing convolutional layers.

In addition to the results for fusing VGG11 models with fixed layer (see Figure 7) we show a direct comparison of partial fusion, clustering and pruning of ensembles with 3 fixed layers in Figure 12.

## B.1. Partial Fusion under different training regimes

The partial fusion results for ResNet18 models reported in Figure 8 use models trained for 200 epochs with SGD (momentum 0.9, learning rate 0.1, weight decay $5 \times 10^{-4}$, batch size 64) and cosine annealing. Under this training regime, full OT Fusion ($\alpha = 0$) already performs close to

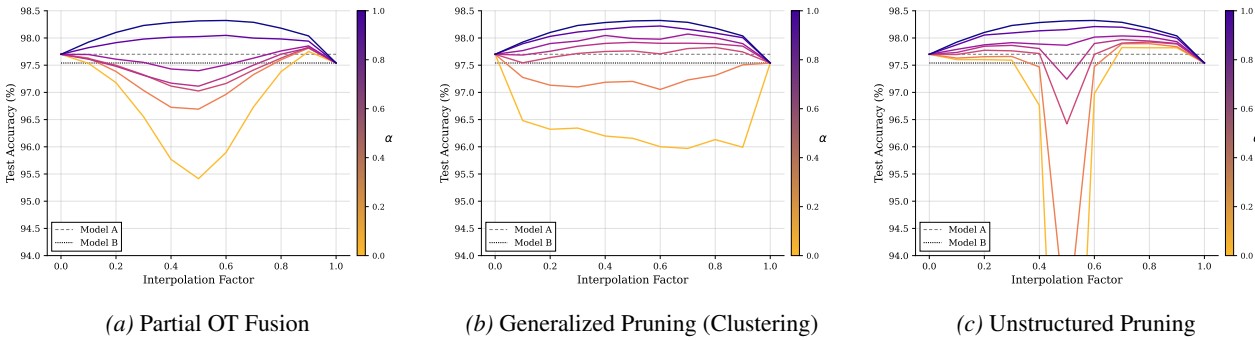

*(a)* Partial OT Fusion       *(b)* Generalized Pruning (Clustering)       *(c)* Unstructured Pruning

*Figure 10.* The same comparison of different methods as shown in Figure 6, with the difference being that the two models A and B are not trained on different parts of MNIST, but both trained until convergence on the whole dataset. Model merging shows less overall efficacy in this setting, especially weight aggregation. The introduced tradeoffs are still often efficient, particularly in the extreme cases when $\alpha$ is near 0 or 1.

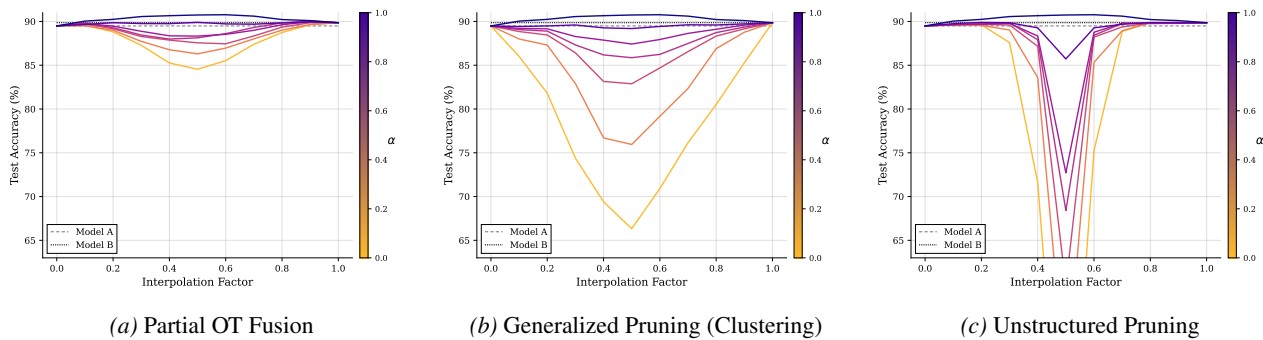

*(a)* Partial OT Fusion       *(b)* Generalized Pruning (Clustering)       *(c)* Unstructured Pruning

*Figure 11.* The same comparison of different methods as shown in Figures 6 and 10, but for CNNs trained independently and identically on the CIFAR10 dataset. As in Singh & Jaggi (2020), pure weight aggregation for CNNs usually does not lead to an increase in accuracy and must be combined with fine-tuning. Nevertheless, we observe interesting features: First, the baseline accuracy for $\alpha = 0$ is surprisingly much higher for OT Fusion compared to generalized pruning, indicating that the objective of Partial OT Fusion seems to be more suitable for CNNs, while clustering is more suitable for MLPs. Further, the efficiency of the parameter/accuracy tradeoff introduced by Partial OT Fusion is less clear for CNNs, but visible again for generalized pruning.

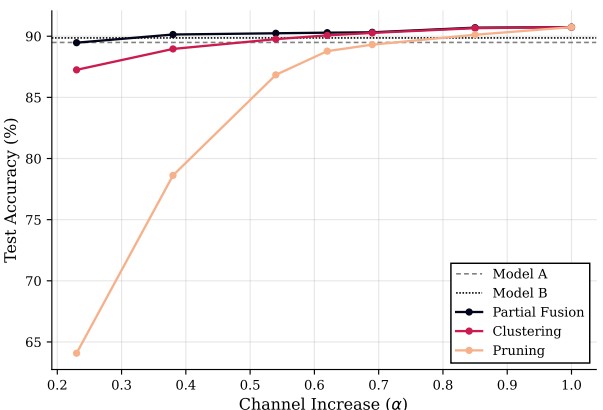

*Figure 12.* This figure presents the respective top lines of Figure 7 from the different panels for better comparison. That is, this figure shows the resulting accuracy of models obtained when all except three layers are partially merged with different fusion methods. The baseline models are two CNNs trained on CIFAR10, which are equally weighted in the aggregation (Interpolation Factor is 0.5).

the individual models.

Figure 13 shows the same partial fusion procedure applied to ResNet18 models trained under a different regime: 300 epochs with SGD (momentum 0.9, learning rate 0.1, weight decay $10^{-4}$, batch size 256) and cosine annealing. Despite achieving similar individual model accuracies ($\approx 94.6\%$), these models are substantially harder to fuse: full OT Fusion drops to roughly 80% at $\lambda = 0.5$, compared to only a minor decrease in Figure 8. Nevertheless, partial fusion continues to interpolate monotonically between OT Fusion and the ensemble, and moderate values of $\alpha$ already recover much of the lost accuracy.

This indicates that the fusibility of models depends not only on architecture and data, but also on training hyperparameters. Partial fusion remains robust in both settings, as it consistently provides a smooth tradeoff regardless of the severity of the fusion barrier.

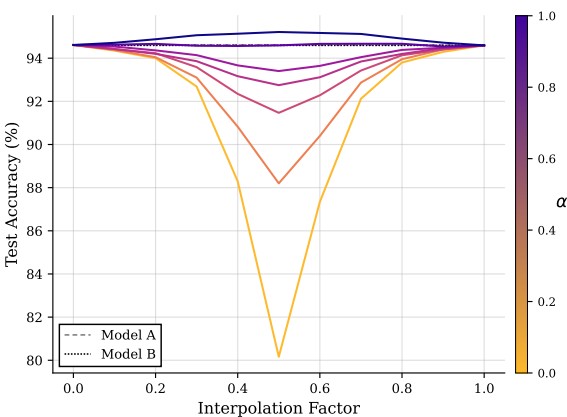

*Figure 13.* Partial fusion of two ResNet18 models trained independently on CIFAR10 with a different training regime than in Figure 8 (300 epochs, weight decay $10^{-4}$). Full OT Fusion ($\alpha = 0$) degrades significantly at intermediate interpolation factors, but partial fusion still interpolates monotonically between weight aggregation and the ensemble. Reported values are averaged over five random seeds.

## B.2. Capacity analysis

We study how the width of MLP models influences the performance of (partial) fusion. We use the same homogeneous setting as in Section 3.3: pairs of MLPs are independently trained on the full MNIST dataset with different random seeds. We vary the hidden dimension across $\{25, 50, 100, 200, 400\}$ (with three hidden layers of equal width in each case). All reported values are averages over five pairs of independently trained models. The results are summarized in Table 2.

**Individual models and naive fusion.** As expected, wider networks lead to better individual model performance, improving from roughly 95% at width 25 to roughly 97.3% at width 400. Naive fusion (averaging weights without any alignment) is catastrophic at the interpolation midpoint $\lambda = 0.5$, dropping to 20.6% for the narrowest networks. While the degradation becomes less severe with increasing width (75.2% at width 200), the behavior is non-monotonic: at width 400, naive fusion drops back to 56.5%.

**Partial OT Fusion.** For all widths, Partial OT Fusion at $\lambda = 0.5$ improves monotonically with $\alpha$, confirming the general trend observed in Section 3.1. However, the relative benefit of leaving neurons isolated (i.e., increasing $\alpha$) is most pronounced for narrow networks. At width 25, the gap between full OT Fusion ($\alpha = 0$, accuracy 64.7%) and the ensemble ($\alpha = 1$, accuracy 95.8%) spans over 31 percentage points. At width 400, this gap shrinks to less than one percentage point (97.2% vs. 97.9%). This suggests that wider layers are inherently easier to align, so there is less need to leave neurons isolated, consistent with observations

in Ainsworth et al. (2023).

We also note that for narrow networks, even moderate values of $\alpha$ yield disproportionate gains. At width 25, going from $\alpha = 0$ to $\alpha = 0.4$ (a 40% increase in neurons per layer) improves accuracy from 64.7% to 85.0% at $\lambda = 0.5$, recovering roughly two thirds of the gap to the ensemble. For wider networks the corresponding improvements are smaller in absolute terms but the fused models are already close to optimal. In particular, at width 200 and above, even full OT Fusion ($\alpha = 0$) at $\lambda = 0.5$ essentially matches the individual models, leaving little room for improvement through partial fusion.

### B.2.1. BASELINES FOR FINE-TUNING

To verify that the improved fine-tuning performance of partially fused models (Table 1) is not merely a consequence of the increased parameter count, we compare against two capacity baselines on both benchmarks. For each value of $\alpha$, we take a single model (model A as it has stronger performance for both datasets) and expand its layers to match the architecture produced by partial fusion at that $\alpha$, using either (i) random weights (*Random widen*) or (ii) the function-preserving Net2WiderNet method (Chen et al., 2016) (*N2W*), and then fine-tune under the same data budget.

Table 3 shows the results. For ResNet-18 on CIFAR-10, random widening destroys the learned representations entirely (chance-level accuracy before fine-tuning), and fine-tuning recovers only $\approx 48\%$ for $\alpha > 0$. Net2WiderNet preserves the original function by construction, and after fine-tuning recovers to $\approx 86.6\%$ regardless of $\alpha$, close to the single-model baseline. For MLPs on MNIST, the picture is qualitatively similar though less dramatic: random widening after fine-tuning slightly degrades with $\alpha$, while Net2WiderNet remains near the single-model baseline. In both settings, the partially fused models improve steadily with $\alpha$ and surpass both capacity baselines for all $\alpha \geq 0.2$. This confirms that the gains from partial fusion stem from the actual aggregation of learned representations, not from the additional capacity alone.

## C. Related literature

We shortly summarize additional closely related literature which was not yet covered in the main part.

Csiszárik et al. (2021) investigate similarity and its relation to an interesting task of transforming the output of one network to be a suitable input to another network. This is similar in spirit to our notion of partial similarity, although our goal of attempting to obtain performance close to that of ensemble models with fewer parameters is quite different to theirs, which is rather to gauge how much performance

*Table 2.* Accuracy (%) on MNIST as a function of hidden layer width for pairs of independently trained MLPs (homogeneous setting as in Section 3.3). Models are fused via Partial OT Fusion for varying $\alpha$. The three columns per width correspond to interpolation factors $\lambda \in \{0, 0.5, 1\}$. All values are averages over five pairs of models.

| | Width 25 | Width 50 | Width 100 | Width 200 | Width 400 |
|---|---|---|---|---|---|
| Model A | 95.2 | 96.2 | 97.0 | 97.1 | 97.2 |
| Model B | 94.9 | 96.4 | 97.1 | 97.1 | 97.4 |
| Naive | 20.6 | 43.0 | 60.6 | 75.2 | 56.5 |
| $\alpha=0$ | 64.7 | 91.0 | 95.8 | 97.0 | 97.2 |
| $\alpha=0.2$ | 81.1 | 94.5 | 96.8 | 97.5 | 97.7 |
| $\alpha=0.4$ | 85.0 | 95.6 | 97.0 | 97.6 | 97.9 |
| $\alpha=0.5$ | 88.8 | 95.6 | 97.2 | 97.6 | 97.9 |
| $\alpha=0.8$ | 94.5 | 96.6 | 97.6 | 97.8 | 97.9 |
| $\alpha=1$ | 95.8 | 96.9 | 97.7 | 97.8 | 97.9 |

*Table 3.* Fine-tuning accuracy (%) for capacity baselines vs. partially fused models. *Random widen* expands a single model with random weights; *N2W* uses Net2WiderNet (Chen et al., 2016) for function-preserving widening. All models are fine-tuned on the same data budget as in Table 1.

| $\alpha =$ | 0.0 | 0.2 | 0.4 | 0.5 | 0.6 | 0.8 | 1.0 |
|---|---|---|---|---|---|---|---|
| **MLP on MNIST (1% of data)** | | | | | | | |
| Random widen | 94.1 | 93.7 | 93.1 | 93.8 | 92.9 | 93.1 | 92.8 |
| N2W | 94.5 | 94.7 | 94.8 | 94.9 | 94.7 | 94.9 | 95.0 |
| Partial fusion | 95.1 | 95.7 | 96.1 | 96.2 | 96.3 | 96.5 | 96.5 |
| *Reference: Model A (no widening): 94.4* | | | | | | | |
| **ResNet-18 on CIFAR-10 (5% of data)** | | | | | | | |
| Random widen | 86.8 | 47.9 | 47.2 | 48.0 | 47.9 | 48.3 | 49.0 |
| N2W | 86.8 | 86.8 | 86.5 | 86.8 | 86.4 | 86.4 | 86.6 |
| Partial fusion | 85.3 | 88.3 | 90.0 | 90.3 | 90.8 | 91.4 | 91.7 |
| *Reference: Model A (no widening): 86.8* | | | | | | | |

is lost when stitching together two networks at different layers.

Measuring similarity of neural networks via certain features of neurons was systematically studied, for instance by Kornblith et al. (2019); Li et al. (2015); Morcos et al. (2018); Raghu et al. (2017). These works primarily focus on comparing information content in layers and thus employ methods such as canonical correlation analysis. While these do yield transformations between the layers, they are often much more structurally complex matrices compared to permutation or stochastic matrices, and thus the effectiveness of using these matrices together with weight averaging is trickier because non-linearities from the activation function play a larger role. Nevertheless, the recent work of Horoi et al. (2024) successfully employed merging of layers using canonical correlation analysis.

Advanced methods using layer transformations are for instance studied in Jordan et al. (2023), who showed that renormalizing activations after alignment significantly improves merged model performance. In this regard also (Yadav et al., 2023) should be mentioned, where among others, a trans-

formation to match varying signs of weights is performed which goes beyond pure permutation or transport matrices. There are also methods to merge models which do not fit the view of transformations between layers: Among others, Matena & Raffel (2022) merges parameters by optimizing a certain posterior motivated by the Fisher information, Ilharco et al. (2023) shows a merging method based on so-called task vectors, and Xu et al. (2024a) merge models within a dual space. In Theus et al. (2025), the authors study linear mode connectivity taking into account more general transformations beyond permutations.

Finally, Weight Space Learning (cf. (Han et al., 2026)) studies general aspects of the space of possible weights of neural networks. We believe our work may serve as a step towards a structured study of varying merging operations (like weight aggregation, ensembles, etc.) on this space.

# D. Details on global optimization with weight-based features

This section provides additional details on Section 2.5.

## D.1. From squared distances to inner products

When comparing neurons based on feature vectors $x$ and $y$, we use the natural objective of minimizing the squared Euclidean distance $\|x - y\|^2$. This can be decomposed as

$$\|x - y\|^2 = \|x\|^2 - 2\langle x, y\rangle + \|y\|^2. \qquad (4)$$

Thus, since in optimal transport only the joint dependence is optimized and marginals are fixed, the problem $\min_{\pi \in \Pi(\mu,\nu)} \int \|x-y\| \, \pi(dx, dy)$ has exactly the same optimizers as $\max_{\pi \in \Pi(\mu,\nu)} \int \langle x, y\rangle \, \pi(dx, dy)$. Thus, one might intuitively set up the layer-wise problems equivalently via maximizing inner products (i.e., maximizing alignment of layers) instead of minimizing distances.

However, in the global problem across layers, the equivalence between minimizing squared costs and maximizing inner products is no longer true, since then the terms $\|x\|^2$

or $\|y\|^2$ may depend on the joint dependence arising from other layers. This is in contrast to the setting of Ainsworth et al. (2023), since therein only permutation matrices $P$ are used as transformations and one always has $\|Px\|^2 = \|x\|^2$, which resolves any issues. In the case of this paper however, general stochastic matrices $K$ often satisfy $\|Kx\|^2 \neq \|x\|^2$. Thus, the two objectives with maximizing inner products or minimizing squared euclidean distances are truly different. In particular, in the important case of weights as features, a linear objective in each fixed-point iteration only arises if we treat each iteration as maximizing inner products, which is detailed in the following. Or, in other words, the translation between these two viewpoints remains as a sub-optimality gap if the fixed-point iterations are applied to the squared objective.

### D.2. Weights as features for normal fusion

We use as features $x_\ell^A = K_{\ell+1}^{A \to B} W_\ell^A$ and $x_\ell^B = W_\ell^B$, so in both cases the outgoing weight matrices of neurons which both map into the world of $B$. The inner product cost matrix is then given by $C_\ell = (x_\ell^A)^T x_\ell^B$. We note that in the case of uniform marginals, the kernels $K_\ell = K_\ell^{A \to B}$ are simply a rescaling by $n_\ell^A$ of the coupling $\pi_\ell$ (and analogously $K_\ell^{B \to A}$ a rescaling by $n_\ell^B$ of $\pi_\ell^T$, so in particular $\frac{1}{n_\ell^B}(K_\ell^{B \to A})^T = \frac{1}{n_\ell^A} K_\ell^{A \to B}$). We can reformulate the global optimization problem of maximizing inner products via the optimization variables $K_\ell$, which, with the reformulation $\int \langle x, y \rangle \, \pi_\ell(dx, dy) = \langle C_\ell, \pi_\ell \rangle_F = \frac{1}{n_\ell^A} \langle K_{\ell+1} W_\ell^A K_\ell^T, W_\ell^B \rangle_F$, leads to (setting $K_0$ and $K_{L+1}$ as the identity matrices)

$$\max_{(K_\ell)_{\ell=1,\ldots,L}} \sum_{\ell=0}^{L} \frac{1}{n_\ell^A} \langle K_{\ell+1} W_\ell^A K_\ell^T, W_\ell^B \rangle_F$$

where $\langle \cdot, \cdot \rangle_F$ is the Frobenius inner product. In particular, the global objective is still linear in each single optimization variable, and hence the fixed point iteration again simply requires the solution of one optimal transport problem in each iteration. In practice we again implement the equivalent problem with squared cost for each step, and further we replace the scaling by $\frac{1}{n_\ell^A}$ by a suitably weight-matrix dependent scaling, so that the contributions to the cost from subsequent layers are roughly even, see the `get_similarity` function in `partial_fusion.py` in the supplemented code. In the algorithm, each iteration runs linearly through the objectives corresponding to the indices $l = 0, \ldots, L$ and we use 10 outer iterations, while we usually observe convergence already after around 6 steps.

When applying this fixed-point algorithm combined with optimal transport to compute the kernels, OT fusion is equivalent to weight-based merging used by Ainsworth et al. (2023) for models with the same number of neurons in each layer, both theoretically and empirically (see experi-

ment `mnist_mlp_git_rebasin_baseline.py`). For our purposes of interpolating between weight aggregation and ensembles, optimal transport is a useful generalization of the optimization over permutations, as it can be used for layers with unequal numbers of neurons and can be extended with partial optimal transport.

### D.3. Extension to partial fusion with weight-based features

We extend the weight-based feature approach to partial fusion. The main difference is that the output space of each layer now has a three-part structure: isolated neurons from $A$, fused neurons (in the world of $B$), and isolated neurons from $B$. Correspondingly, we need transformations $K_\ell^{A \to F}$ and $K_\ell^{B \to F}$ mapping neurons of $A$ and $B$ into this joint space.

These transformations have block structures reflecting the partition. With columns indexed by $(I_\ell^A, F_\ell^A)$ and rows by $(I_\ell^A, F_\ell^B, I_\ell^B)$:

$$K_\ell^{A \to F} = \begin{bmatrix} \mathbb{1}_{I_\ell^A} & 0 \\ 0 & K_\ell^{A \to B}[F^B, F^A] \\ 0 & 0 \end{bmatrix},$$

$$K_\ell^{B \to F} = \begin{bmatrix} 0 & 0 \\ \mathbb{1}_{F_\ell^B} & 0 \\ 0 & \mathbb{1}_{I_\ell^B} \end{bmatrix},$$

where $K_\ell^{A \to B}[F^B, F^A]$ denotes the restriction of $K_\ell^{A \to B}$ to fused neurons. The outgoing-weight-based features for layer $\ell$ are then

$$x_\ell^A = K_{\ell+1}^{A \to F} W_\ell^A, \qquad x_\ell^B = K_{\ell+1}^{B \to F} W_\ell^B,$$

which both map into the joint output space and can thus be compared directly. With this definition, we can now follow the steps as in normal fusion, again taking as optimization variable the kernel $K_{\ell+1}^{A \to F}$. We similarly find that the global optimization remains linear and is again a partial optimal transport problem as the layer-wise problem. For details, we refer to the function `compute_kernels_pcd` in `base_fusion.py` in the supplemented code.

## E. Extension of partial fusion for CNNs

Applying partial fusion to combine convolutional layers is achieved by replacing neurons in a MLP model by channels. One key difference is the dimension of the support for a channel. We simply flatten the weights or activations of a channel to a vector and compute the cost between different channels, typically the squared euclidean norm of the vectors.

The "flatten" operation often used in CNNs to transition from convolutional to fully connected layers needs special

attention in partial fusion. When flattening the activations the number of channels output from the convolutional layer does not correspond to the amount of neurons of the fully connected layer. This means also the respective kernel does not match the dimension of the layers. We extend the smaller kernel that matches the channels, to match the number of neurons. Let every channel be flattened to $r$ neurons, then we construct matrix $D_{ij}$ for every value $k_{ij}$ in the kernel.

$$D_{ij} = k_{ij}\mathbf{I}_r = \begin{bmatrix} k_{ij} & 0 & \cdots & 0 \\ 0 & k_{ij} & \cdots & 0 \\ \vdots & \vdots & \ddots & \vdots \\ 0 & 0 & \cdots & k_{ij} \end{bmatrix},$$

where $\mathbf{I}_r$ is the $r \times r$ identity matrix, and thus $D_{ij}$ is a $r \times r$ diagonal matrix. Then we expand the kernel to $K_{\text{expanded}}$ which matches the dimension of the weight matrix of the fully connected layer,

$$K_{\text{expanded}} = \begin{bmatrix} D_{11} & D_{12} & \cdots & D_{1m} \\ D_{21} & D_{22} & \cdots & D_{2m} \\ \vdots & \vdots & \ddots & \vdots \\ D_{m1} & D_{m2} & \cdots & D_{mm} \end{bmatrix}.$$

### E.1. Extension of partial fusion for ResNets

Residual neural networks (He et al., 2016) modify standard CNN models by introducing residual connections and batch normalization. These two additions to the model architecture require special attention when (partially) fusing models. We describe the necessary modifications using the standard fusion notation with kernels $K_\ell^{A \to B}$ and $K_\ell^{B \to A}$; the extension to partial fusion follows directly by replacing these with the partial fusion kernels $K_\ell^{A \to F}$ and $K_\ell^{B \to F}$ as described in Section D.3.

**Identity shortcut.** Standard residual connections add the input of a layer block directly to its output, i.e., $\mathbf{y} = \mathcal{F}(\mathbf{x}) + \mathbf{x}$. Because the identity-mapped input $\mathbf{x}$ is added element-wise to the block's output, the channel ordering at the end of the block is tied to the channel ordering at its input. Consider a residual block whose first convolutional layer is at position $\ell_1$ and whose last convolutional layer is at position $\ell_r$. The skip connection identifies channels at position $\ell_1$ (the block input) with channels at position $\ell_r + 1$ (the block output). The element-wise addition therefore requires that the kernels at these two positions coincide:

$$K_{\ell_1}^{A \to B} = K_{\ell_r+1}^{A \to B}. \tag{5}$$

We enforce this by solving a single shared (partial) optimal transport problem whose features aggregate information from both ends of the block. Following Section D.2, the outgoing-weight features for layer $\ell$ are $x_\ell^A = K_{\ell+1}^{A \to B} W_\ell^A$

and $x_\ell^B = W_\ell^B$. To tie the kernel computation for the first and last layer of the block, we aggregate the cost matrices of both layers into a single shared optimal transport problem:

$$\tilde{C} = C_{\ell_1} + C_{\ell_r+1}. \tag{6}$$

Here we use the fact that for an identity shortcut all layers in the block preserve the number of channels, so that the columns of $x_{\ell_1}$ and $x_{\ell_r}$ are indexed by the same set of channels and the cost matrices have the same dimension. Solving the resulting (partial) optimal transport problem

$$\tilde{\pi} = \underset{\pi \in \Pi(\mu_l^A, \mu_l^B)}{\operatorname{argmin}} \langle \tilde{C}, \pi \rangle_F$$

produces a shared kernel $K_{\text{shared}}^{A \to B}$, which is set as

$$K_{\ell_1}^{A \to B} := K_{\text{shared}}^{A \to B}, \qquad K_{\ell_r+1}^{A \to B} := K_{\text{shared}}^{A \to B}.$$

This shared kernel is used to transform both the weights at the block boundaries and the activations passed through the residual connection. The kernels for any intermediate layers $\ell_{1+1}, \ldots, \ell_r$ within the block are computed independently.

**Projected shortcut.** Commonly used residual networks like ResNet18 do not only use identity shortcuts. When the residual block involves downsampling or a change in the number of channels, the identity shortcut is replaced by a learned $1 \times 1$ convolution: $\mathbf{y} = \mathcal{F}(\mathbf{x}) + W_s\mathbf{x}$. Since $W_s$ maps between potentially different channel dimensions, the channel orderings at positions $\ell_1$ and $\ell_r + 1$ are no longer tied, and the constraint (5) is dropped. The kernels $K_{\ell_1}^{A \to B}$ and $K_{\ell_r+1}^{A \to B}$ are computed independently. The projection weights are then transformed analogously to standard weight matrices (cf. Section 2.2), using the forward kernel at the output position and the backward kernel at the input position:

$$\widetilde{W}_s^A = K_{\ell_r+1}^{A \to B} W_s^A K_{\ell_1}^{B \to A}. \tag{7}$$

**Batch normalization.** Batch normalization layers are ignored when computing kernels. Each batch normalization layer following convolutional layer $\ell$ has learned affine parameters $\gamma_\ell$ (scale) and $\beta_\ell$ (shift), which are vectors indexed by the channels at position $\ell + 1$. These are merged (or kept isolated) by applying the forward kernel of the preceding layer:

$$\hat{\gamma}_\ell^A = K_{\ell+1}^{A \to B} \gamma_\ell^A, \qquad \hat{\beta}_\ell^A = K_{\ell+1}^{A \to B} \beta_\ell^A, \tag{8}$$

and analogously for model $B$ (where no transformation is needed since the fused network lives in the world of $B$). The running statistics (mean $\mu_{\text{BN}}$ and variance $\sigma_{\text{BN}}^2$) cannot be merged linearly due to their nonlinear dependence on the activations. Instead, they are recomputed by running a forward pass through the fused model using a subset of the training dataset.

If no data at all is available for recalibration, there is a simple but elegant heuristic one can use instead. To this end, as our fused model is considered as part of the world of model $B$, we need the matrix $\widehat{W}_l^A := W_l^A K_l^{B \to A}$. Assuming the input neurons (so the values of the neurons of layer $l$ in the world of model $B$) are standard Gaussian, then a quick calculation shows that one can compute the correlation of neuron $i$ in network $A$ and neuron $j$ in network $B$ as the cosine similarity $\angle(\widehat{W}_A[i,:], W_B[j,:])$ of the weight matrices $W_A[i,:]$ and $W_B[j,:]$. Thus, with the notation as in (Jordan et al., 2023), for a merged neuron $X_\alpha = (1-\alpha)X_1 + \alpha X_2$ (where $X_1$ is the $i$-th neuron of net $A$ and $X_2$ the $j$-th neuron of net $B$), one obtains the batch norm statistics by linearity for the mean, and by the formula $\mathrm{Var}(X_\alpha) = (1-\alpha)^2\mathrm{Var}(X_1) + \alpha^2\mathrm{Var}(X_2) + 2\alpha(1-\alpha)\mathrm{std}(X_1)\mathrm{std}(X_2) \cdot \angle(\widehat{W}_A[i,:], W_B[j,:])$. This approach was used to fuse the ResNet models in the model aggregation and fine-tuning setting described in Section 3.2, for the code implementation see function `_fill_bn_aligned` in `fusion_model.py`.

## F. Interpolation with clustering and pruning

In Section 3 we use generalized clustering and unstructured pruning of ensembles to interpolate between models, with differing interpolation factors $\lambda$. This is achieved by suitably scaling the neurons of the respective models within the ensemble. Concretely for clustering an ensemble $E$ given as $(1 - \lambda)A + \lambda B$, the probabilities $\mu_\ell^E$ for clustering of the neurons are not uniform, but scaled by $(1 - \lambda)$ for neurons of Model $A$, and $\lambda$ for neurons of Model $B$ (and finally normalized to one again). Similarly, for unstructured pruning the assigned importance of neurons arising from Model $A$ are multiplied by $(1 - \lambda)$ and neurons from Model $B$ by $\lambda$. Note the double role of $\lambda$ for both the weights in the output layer and for the importance of the neurons.

## G. Partial fusion as generalized pruning of the ensemble

We shortly show that this generalized pruning of ensembles can be regarded as a more general way of combining two networks compared to the partial fusion method described in Section 2.2. Indeed, if $E$ is the ensemble of $A$ and $B$, that is, if $W_\ell^E = \begin{bmatrix} W_\ell^A & 0 \\ 0 & W_\ell^B \end{bmatrix}$ then $S$ corresponds to the partial fusion of $A$ and $B$ with the choice

$$K_\ell^{E \to S} = \begin{bmatrix} \mathbb{1} & 0 & 0 & 0 \\ 0 & \lambda K_\ell^{A \to B} & (1 - \lambda)\mathbb{1} & 0 \\ 0 & 0 & 0 & \mathbb{1} \end{bmatrix},$$

$$K_\ell^{S \to E} = \begin{bmatrix} \mathbb{1} & 0 & 0 \\ 0 & K_\ell^{B \to A} & 0 \\ 0 & \mathbb{1} & 0 \\ 0 & 0 & \mathbb{1} \end{bmatrix},$$

where the four blocks of $E$ correspond to $I_\ell^A, F_\ell^A, F_\ell^B, I_\ell^B$ and the three blocks of $S$ to $I_\ell^A, F_\ell^B, I_\ell^B$, which are the occurring indices in the fused model. To be precise, for simplicity, we hereby assume that the neurons of $A$ and $B$ are already properly sorted in such a way that the block structure $W_\ell^E = \begin{bmatrix} W_\ell^A & 0 \\ 0 & W_\ell^B \end{bmatrix}$ is consistent with this split; that is, that the first neurons of $A$ are the ones in $I_\ell^A$, and the last neurons of $B$ are the ones in $I_\ell^B$.

## H. Parameter counts for generalized pruning

We parametrize the generalized pruning methods by the increase of neurons per layer ($\alpha$) compared to the size of the initial networks (assuming similar sizes). In this section we show the resulting parameter counts depending on $\alpha$ for two feedforward networks with the same layer sizes. We first examine upper bounds for the parameters and then show the parameter counts in concrete examples.

Let $W \in \mathbb{R}^{m \times n}$ be the weight matrix for a layer with $n$ input neurons and $m$ output neurons. When applying generalized pruning to an ensemble we increase the amount of neurons by a factor of $1 + \alpha$. Then the weight matrix has $(1+\alpha)^2 nm$ entries. We are interested in the parameter count excluding block matrices in $W$ that are entirely zeros.

*Table 4.* Number of non-zero parameters of the weight matrix between layers $\ell$ and $\ell + 1$ for a fixed fraction $\alpha$ of retained neurons in both layers, when the initial number of neurons in the layers is $n$ and $m$. Pruning and clustering counts depend on the specific network configuration (i.e., the specific neurons which are deleted by pruning and the specific optimal clusters), while partial fusion yields a fixed count.

| Method | Best Case | Worst Case |
|---|---|---|
| Pruning | $2\left(\frac{1+\alpha}{2}\right)^2 nm$ | $(1 + \alpha^2)\,nm$ |
| Clustering | $2\left(\frac{1+\alpha}{2}\right)^2 nm$ | $(1 - \alpha^2 + 2\alpha)\,nm$ |
| Partial Fusion | $(1 - \alpha^2 + 2\alpha)\,nm$ | |

The theoretic bounds on the parameters (see Table 4) show that pruning is more parameter efficient than partial fusion. Clustering can vary to have the same amount of parameters as pruning or partial fusion. The best case scenario for pruning corresponds to pruning the same amount of neurons from both models and the worst case corresponds to pruning only neurons from one of the two models. For clustering the best case scenario is to always cluster neurons from the same network together. The worst case is to always cluster pairs of neurons from both networks together, like partial fusion does.

Table 5 shows the increase in parameters for a concrete example for $\alpha = 0.5$. It shows the clear difference between pruning and partial fusion and the large variability in the

*Table 5.* A concrete example of the parameter counts resulting from Table 4 for $\alpha = 0.5$. Pruning needs the least amount of parameters, as it has no cross connections between the two models (for $\alpha = 0$ pruning leads to half the parameters of a single model). The parameter count for clustering can vary a lot depending on which neurons are combined.

| Method | Best Case | Worst Case |
|---|---|---|
| Pruning | 1.125 | 1.25 |
| Clustering | 1.125 | 1.75 |
| Partial Fusion | | 1.75 |

parameter count for clustering.

We also note that in concrete models the increase in (non-zero) parameters is overall lower, as the number of input and output neurons stays consistent between a single model and the pruned ensemble, i.e., we always have $\alpha = 0$ for input and output layers.

## I. Storage, memory and wall-clock times

This section provides a short discussion how the benefits in terms of effective parameter counts translate to practical benefits in terms of storage, memory, and wall-clock times. Throughout, we differentiate two regimes: Regime 1 for smaller models, where it is more efficient to consider partially fused models as dense models including zeroes. And Regime 2 for larger models, where it is more efficient to consider each non-zero block as a parallel layer.

Firstly, storage cost relates practically one-to-one to the number of effective parameters. This is the case even in the Regime 1 by only reconstructing the dense matrix at load time.

The discussion of working memory is the most subtle. Broadly speaking, two main components contribute to memory: model parameters and activations. In theory, ensembles can be very efficient regarding memory, as they can be evaluated sequentially, and only the output needs to be kept in memory simultaneously to be aggregated eventually. However, in practical pipelines, particularly if models should be evaluated many times or autoregressively, such a sequential evaluation is often infeasible as it would lead to increases in latency by orders of magnitude. So in practical settings, ensembles of two equally sized models often really do incur twice the memory cost, but it would clearly be wrong to state it as a necessity. For partially fused models, while the memory contribution in terms of parameters is as governed by the effective parameters (in Regime 2), the memory requirements for the activations are dictated by the layer width, that is, the parameter $\alpha$. So roughly speaking, partially fusing two equally sized networks with parameter $\alpha = 0.5$ would use memory inbetween a factor of $1.5\times$ (as for activations)

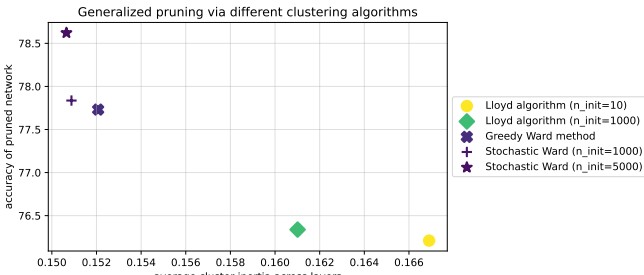

*Figure 14.* Comparison of different clustering algorithms for generalized pruning of a feed forward network to $0.4\times$ its original size, averaged over 10 random seeds. While variance is relatively high (standard deviations across random seeds was around 2% accuracy for each point in the figure), the trend seen in this figure is quite representative for all the experiments we ran with different clustering algorithms: Better clustering performance correlates with higher accuracy, and hierarchical clustering outperforms Lloyd's algorithm, even if the latter uses many random initializations. Notably, the ordering of the run times was, from fastest to slowest: Greedy Ward, Lloyd ($n_{init} = 10$), Stochastic Ward ($n_{init} = 1000$), Stochastic Ward ($n_{init} = 5000$) and Lloyd ($n_{init} = 1000$).

and a factor of $1.75\times$ (as for effective parameters) of an individual model.

Regarding Wall-Clock-Time: For a fixed architecture and input size, FLOPs scale pretty much linearly with the number of parameters in a network as $\alpha$ varies, and hence in Regime 2, with the number of effective parameters. Therefore, for instance with (and assuming equally sized layers), a partially fused network uses $1.75\times$ the FLOPs of a single model, while ensembles use $2\times$. With modern implementations on GPU however, theoretical FLOP calculations don't immediately transfer to Wall-Clock-Times, especially for small models. While a thorough analysis of the best ways to utilize the reduced parameter counts of partially fused models compared to ensembles would go beyond the scope of the paper, we report in Table 6 a numerical comparison. This shows that at least for large models (i.e., wide layers), the benefits in terms of Wall-Clock-Time are roughly as predicted by the FLOP calculation.

## J. On the choice of clustering algorithm

Recall the clustering objective

$$\pi_\ell^{E,S} = \underset{\pi_\ell \in \Pi(\mu_\ell^E, *_m)}{\operatorname{argmin}} \int \|x - y\|^2 \, \pi(dx, dy),$$

where $\mu_\ell^E$ determines the weighted sets of points to cluster and $m$ is the number of cluster centers. In the regime relevant for pruning, usually $m$ is a significant fraction, usually between $10\% - 90\%$, of the total number of points. Whereas in most applications of K-means clustering, there are far fewer cluster centers compared to points. Further, any clustering algorithm is basically a heuristic, because solving

*Table 6.* Wall-clock time relative to single model, $\alpha = 0.5$, batch size 64. Median of 5000 runs. Regime 1 refers to a dense implementation including zeroes in the layer, while Regime 2 refers to an implementation where non-zero blocks are treated as parallel layers. Widths refer to the single-model hidden dimension of MLPs with three hidden layers; the fused width is $1.5\times$ this value. While noisy, for large models the wall clock time roughly approaches the theoretical values predicted by the FLOPs calculation. Measurements use CUDA events (using `torch.cuda.Event(enable_timing=True)`) on an NVIDIA RTX A6000 (48GB), PyTorch 2.4.1, CUDA 12.4.0.

| Width | Single (ms) | Ensemble | Regime 1 | Regime 2 | FLOPs R1 | FLOPs R2 |
|---|---|---|---|---|---|---|
| 100 | 0.057 | 2.08× | 0.95× | 2.45× | 2.25× | 1.75× |
| 500 | 0.061 | 2.11× | 1.03× | 2.50× | 2.25× | 1.75× |
| 3000 | 0.231 | 2.01× | 2.03× | 2.24× | 2.25× | 1.75× |
| 8000 | 1.249 | 2.02× | 2.24× | 1.69× | 2.25× | 1.75× |
| 12000 | 2.854 | 2.02× | 1.96× | 1.59× | 2.25× | 1.75× |
| 16000 | 4.878 | 2.00× | 2.27× | 1.81× | 2.25× | 1.75× |

the clustering problem globally is NP-hard (Aloise et al., 2009). We stress this point because the standard clustering algorithm, Lloyd's algorithm (Lloyd, 1982), should be seen as a good heuristic aimed at efficiency for large datasets and relatively few cluster centers, which is precisely not the regime which occurs here. Comparatively, for smaller datasets and larger number of cluster centers, hierarchical clustering algorithms are known to more often obtain near-optimal solutions, albeit at a potential increase in runtime (Jain et al., 1999).

For our experiments, we employ a variant of Ward's method as the hierarchical clustering algorithm (Ward Jr., 1963). The standard Ward's method is a greedy algorithm: Each point is initialized as being its own cluster and in each step, two clusters are combined so that the overall inter-cluster variance is increased the least. To improve cluster quality, we employ a stochastic variant, which randomly chooses two clusters to combine (via softmax weighting in that the greedy choice is still chosen as the likeliest option), and then use many restarts of the algorithm (in the main experiments, usually 50000). See Figure 14 for a rough illustration of the effect of different clustering algorithms for generalized pruning.

We shortly emphasize that also some other model merging methods can formally be regarded as a generalized pruning method, while the respective algorithms therein can be seen as particular clustering algorithms. For instance (Stoica et al., 2024) clusters neurons of the ensemble into pairs of two which have the most correlated activations. With a slight extrapolation of their method, we can also only cluster the most similar neurons into pairs of two and keep the rest isolated, to make it a variable size clustering. We emphasize that the scope of (Stoica et al., 2024) is of course much wider (merging multiple models trained on different tasks), but we show how simply plugging their approach into the generalized pruning methods of this paper lead to reasonable results, see Figure 15.

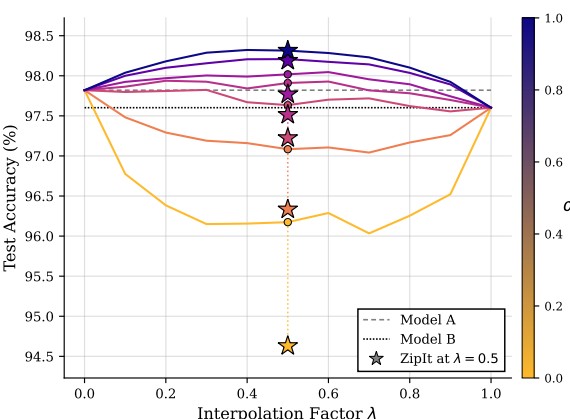

*Figure 15.* MLP merging on the MNIST dataset as in Figure 10. The figure shows that the method in (Stoica et al., 2024), applied to two models, can formally be regarded as a particular clustering algorithm and used within the proposed generalized pruning framework. As the method is aimed at equally weighted merges, we plot the respective values for $\lambda = 0.5$, which are comparable but slightly lower compared to the plotted lines which arise from hierarchical clustering.

## K. On the choice of $\lambda$

While the role of $\lambda \in [0, 1]$ is generally easy to understand (it assigns importance to the individual networks), it is much less clear how to optimally choose $\lambda$ in practice. In this paper, we do not intend to solve this question, and instead our goal is to show benefits for varying $\lambda$ in order to showcase that the introduced methods are robust across different practical procedures for choosing $\lambda$. Importantly, even performance for $\lambda$ in which the merged model performs worse than both individual networks can be important, as benefits on downstream tasks may not always correlate with performance directly after merging. A slight extension of the fine-tuning experiment reported in Table 1 can provide insight here: While performance before fine-tuning is often best for $\lambda = 0.1$, the performance after fine-tuning is consistently better for $\lambda = 0.5$, see Table 7.

We further emphasize that generally just always choosing whatever $\lambda$ works best in downward tasks is also not always a feasible procedure, as in realistic scenarios a test set to decide which $\lambda$ works best may not be available.

## L. Differences in neuron-level similarities

In this section we summarize some observations about differences in neuron-level similarities across networks which can motivate and explain certain parts of the partial fusion methods introduced in this paper. First, we observe that when leaving least-similar neurons isolated, this (significantly) improves the average similarity of non-isolated neurons. And second, we also illustrate differences between neuron-level similarity across layers and across the MLP and CNN examples.

Figures 16 and 17 showcase neuron-level similarities within fixed layers for two pairs of MLPs and CNNs (trained homogeneously as in Figures 10 and 11). We use activation vectors as features for neurons (since this is simpler than weight matrices, which would need to be aligned across layers) arising from 1000 data points, as in the main experiments. For each neuron, we look at its nearest neighbor distance, which is the distance to the most similar other neuron, and its average distance to all other considered neurons. We evaluate those both within the same network and across different networks. So for instance, the nearest neighbor distance across the network measures, for each neuron, the (euclidean) distance to the most similar neuron in the other network in the same layer. In the figures, one can see the variation of these distances across neurons. We see that, especially for the nearest neighbor distance, there definitely are some large variations in both extremes, so there are unusually similar and unusually dissimilar neurons in most layers. This provides the motivation for our proposed methods since it suggests that adjusting either ensembles (by

merging most similar neurons) or weight aggregation (by not merging most isolated ones) can lead to good tradeoffs.

We make the change by leaving the 20% most dissimilar neurons isolated more explicit in Tables 8 and 9. These tables report the average values over the histograms (the *All neurons* part), and then the conditional average over the 80% of smallest values for each histogram. We see that for CNNs and MLPs alike, significant differences arise.

In Tables 10 and 11 we showcase another interesting observation which may partly explain the inductive bias of Partial OT Fusion (discussed in Section 2.3 related to the relative performances of the different methods observed in Figures 10 and 11). In particular, we find that for CNNs, there is a fairly significant difference of nearest neighbor distances within versus across networks. Particularly in early layers, the cross-network distances are smaller than the within-network distances, suggesting that many channels really satisfy a unique role in the network, and a channel in the other network has a similar role. Of course, the *measured* similarities based on activations can never fully explain the effects of the inductive *bias* of the Partial OT Fusion method, which must go beyond the data at hand. In this regard, the tables simply give an indication that such patterns may arise, and these patterns may be different across MLPs, CNNs and across layers, hence it is fundamentally valuable to have different tools available which can be combined flexibly to aggregate models. In particular the patterns that for early layers, cross-network similarities are more similar (perhaps since the same basic functionalities arise in both networks) and that those cross-network similarities deteriorate in later layers (perhaps since the input to the deeper layers, which is an increasingly divergent output of previous layers prevents the exact same functional roles from arising) may be universal and exploited to navigate the best possible merging techniques. That is, if more similarities within-network are expected, use generalized pruning methods, and if more similarities across networks are expected, use partial fusion.

*Table 7.* Accuracy (%) on CIFAR-10 (ResNet-18) before and after fine-tuning with 5% of the training data, comparing fusion at $\lambda = 0.1$ and $\lambda = 0.5$. Before fine-tuning, $\lambda = 0.1$ substantially outperforms $\lambda = 0.5$ at low $\alpha$ (e.g. 77.04 vs. 66.39 at $\alpha = 0.0$), since the fused model stays close to one parent. However, after fine-tuning this advantage vanishes: $\lambda = 0.5$ consistently surpasses $\lambda = 0.1$ across all $\alpha$, with the gap being most pronounced at small $\alpha$ (e.g. +2.71 at $\alpha = 0.4$). This indicates that merging accuracy is not a reliable predictor of downstream performance after fine-tuning.

|  | $\alpha = 0.0$ | $\alpha = 0.2$ | $\alpha = 0.4$ | $\alpha = 0.5$ | $\alpha = 0.6$ | $\alpha = 0.8$ | $\alpha = 1.0$ |
|---|---|---|---|---|---|---|---|
| $\lambda = 0.1$ | | | | | | | |
| Fusion | **77.04** | **77.41** | 77.52 | 77.69 | 78.46 | 78.84 | 79.11 |
| Fine-tuning | 85.31 | 85.79 | 87.30 | 87.99 | 88.61 | 89.42 | 89.64 |
| $\lambda = 0.5$ | | | | | | | |
| Fusion | 66.39 | 77.19 | **83.78** | **87.35** | **88.92** | **90.58** | **91.29** |
| Fine-tuning | **85.32** | **88.31** | **90.01** | **90.28** | **90.81** | **91.39** | **91.81** |

*Table 8.* Neuron similarity statistics for two MLPs trained on the same data (MNIST) with different random seeds. The values are averages across neurons of a given layer in a given network. Similarities are either measured within a given network or across two networks. The top shows the averages across all neurons, which are the integrated values of the histograms shown in Figure 16. The middle shows the averages conditioned on the 80% least isolated/most similar neurons (so conditioned on the left part of the histogram values in Figure 16). The bottom shows the resulting difference.

| | **Nearest Neighbor Distance** | | | | **Mean Distance to All** | | | |
|---|---|---|---|---|---|---|---|---|
| | Within | | Across | | Within | | Across | |
| Layer | A | B | A→B | B→A | A | B | A→B | B→A |
| *All neurons* | | | | | | | | |
| 1 | 33.8 | 34.4 | 32.3 | 32.3 | 58.7 | 60.6 | 59.4 | 59.4 |
| 2 | 26.7 | 27.1 | 27.4 | 26.5 | 55.0 | 58.2 | 56.5 | 56.5 |
| 3 | 26.1 | 27.4 | 26.8 | 28.4 | 60.4 | 62.9 | 61.7 | 61.7 |
| *80% least isolated neurons* | | | | | | | | |
| 1 | 30.7 | 30.9 | 29.2 | 29.3 | 54.6 | 56.0 | 55.6 | 54.6 |
| 2 | 24.1 | 23.6 | 24.6 | 23.0 | 51.6 | 54.1 | 53.1 | 52.4 |
| 3 | 22.7 | 24.3 | 23.7 | 25.2 | 56.6 | 59.6 | 58.2 | 57.9 |
| *Difference (isolated neuron contribution)* | | | | | | | | |
| 1 | 3.1 | 3.5 | 3.1 | 3.0 | 4.0 | 4.6 | 3.8 | 4.8 |
| 2 | 2.6 | 3.5 | 2.8 | 3.5 | 3.4 | 4.1 | 3.4 | 4.1 |
| 3 | 3.3 | 3.1 | 3.1 | 3.2 | 3.8 | 3.3 | 3.5 | 3.8 |

*Table 9.* The same neuron similarity statistics as in Table 8, but for two CNN models instead of MLPs, also trained on the same data (CIFAR10) with different random seeds. We note the overall difference in magnitude of differences compared to Table 8, and also the larger relative difference of nearest neighbor distances to mean distances, most notably in the first layer.

| | Nearest Neighbor Distance | | | | Mean Distance to All | | | |
|---|---|---|---|---|---|---|---|---|
| | Within | | Across | | Within | | Across | |
| Layer | A | B | A→B | B→A | A | B | A→B | B→A |
| *All channels* | | | | | | | | |
| 1 | 2.31 | 2.35 | 1.68 | 1.76 | 7.56 | 7.39 | 7.40 | 7.40 |
| 2 | 3.81 | 3.85 | 3.43 | 3.43 | 8.71 | 8.62 | 8.63 | 8.63 |
| 3 | 4.23 | 4.24 | 4.03 | 4.04 | 7.49 | 7.42 | 7.44 | 7.44 |
| 4 | 1.89 | 1.88 | 1.79 | 1.79 | 2.91 | 2.91 | 2.91 | 2.91 |
| 5 | 1.73 | 1.72 | 1.72 | 1.75 | 3.35 | 3.47 | 3.41 | 3.41 |
| 6 | 1.12 | 1.21 | 1.25 | 1.30 | 3.76 | 3.80 | 3.80 | 3.80 |
| 7 | 2.26 | 2.32 | 2.63 | 2.74 | 8.94 | 9.07 | 9.06 | 9.06 |
| 8 | 3.62 | 3.56 | 4.27 | 4.34 | 13.44 | 13.52 | 13.50 | 13.50 |
| *80% least isolated channels* | | | | | | | | |
| 1 | 1.47 | 1.47 | 1.11 | 1.16 | 6.04 | 5.93 | 5.90 | 5.94 |
| 2 | 2.99 | 2.94 | 2.73 | 2.79 | 7.48 | 7.48 | 7.41 | 7.49 |
| 3 | 3.95 | 3.96 | 3.75 | 3.76 | 7.09 | 7.03 | 7.05 | 7.04 |
| 4 | 1.66 | 1.64 | 1.57 | 1.58 | 2.66 | 2.66 | 2.66 | 2.66 |
| 5 | 1.56 | 1.51 | 1.55 | 1.54 | 3.01 | 3.06 | 3.07 | 2.99 |
| 6 | 0.95 | 0.97 | 1.06 | 1.06 | 3.29 | 3.30 | 3.31 | 3.30 |
| 7 | 1.88 | 1.90 | 2.20 | 2.26 | 8.02 | 8.07 | 8.16 | 8.04 |
| 8 | 2.93 | 2.90 | 3.55 | 3.66 | 12.09 | 12.05 | 12.16 | 12.02 |
| *Difference (isolated channel contribution)* | | | | | | | | |
| 1 | 0.83 | 0.88 | 0.57 | 0.60 | 1.52 | 1.47 | 1.49 | 1.46 |
| 2 | 0.82 | 0.92 | 0.70 | 0.63 | 1.23 | 1.14 | 1.22 | 1.14 |
| 3 | 0.28 | 0.29 | 0.28 | 0.28 | 0.40 | 0.39 | 0.39 | 0.40 |
| 4 | 0.23 | 0.24 | 0.22 | 0.22 | 0.25 | 0.25 | 0.25 | 0.25 |
| 5 | 0.17 | 0.21 | 0.17 | 0.20 | 0.34 | 0.41 | 0.34 | 0.42 |
| 6 | 0.18 | 0.24 | 0.19 | 0.24 | 0.48 | 0.51 | 0.49 | 0.49 |
| 7 | 0.37 | 0.42 | 0.43 | 0.49 | 0.92 | 1.01 | 0.90 | 1.02 |
| 8 | 0.69 | 0.66 | 0.72 | 0.69 | 1.35 | 1.47 | 1.35 | 1.48 |

*Table 10.* Nearest-neighbor distances for MLPs as in Table 8. The table emphasizes the (in the case of MLPs very weak) pattern that cross-network similarities are lower in earlier layers than within-network similarities.

| Layer | Within A | Within B | Cross A→B | Pattern |
|---|---|---|---|---|
| 1 | 33.8 | 34.4 | 32.3 | cross < within |
| 2 | 26.7 | 27.1 | 27.4 | cross ≈ within |
| 3 | 26.1 | 27.4 | 26.8 | cross ≈ within |

*Table 11.* Nearest-neighbor distances of CNNs as in Table 9. The table shows that early layers show lower cross-network similarities compared to within-network similarities, indicating unique roles of certain channels in early layers which are present in both networks. The pattern reverses in deeper layers, but to a lesser degree.

| Layer | Channels | Within A | Within B | Cross A→B | Pattern |
|---|---|---|---|---|---|
| 1 | 64 | 2.31 | 2.35 | 1.68 | cross < within |
| 2 | 128 | 3.81 | 3.85 | 3.43 | cross < within |
| 3 | 256 | 4.23 | 4.24 | 4.03 | cross < within |
| 4 | 256 | 1.89 | 1.88 | 1.79 | cross < within |
| 5 | 512 | 1.73 | 1.72 | 1.72 | cross ≈ within |
| 6 | 512 | 1.12 | 1.21 | 1.25 | cross > within |
| 7 | 512 | 2.26 | 2.32 | 2.63 | cross > within |
| 8 | 512 | 3.62 | 3.56 | 4.27 | cross > within |

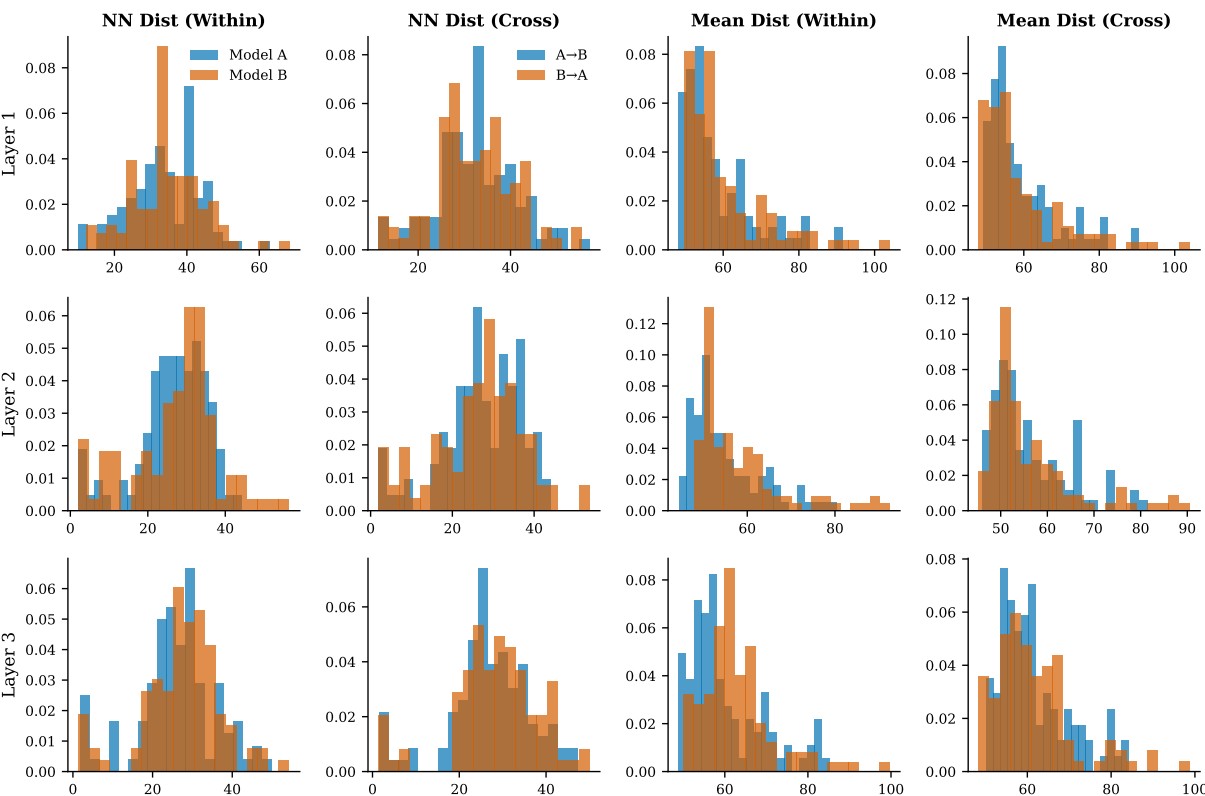

*Figure 16.* Neuron distance distributions for two MLPs trained on the same data with different random seeds (as in Table 8). The nearest neighbor distances (NN Dist) are, for each neuron, the distance to the most similar other neuron, either within the same network (Within) or across the other network (Cross). Mean distances are the average distance, of each neuron, to all other neurons. Distances are measured in euclidean distance of the activation vectors.

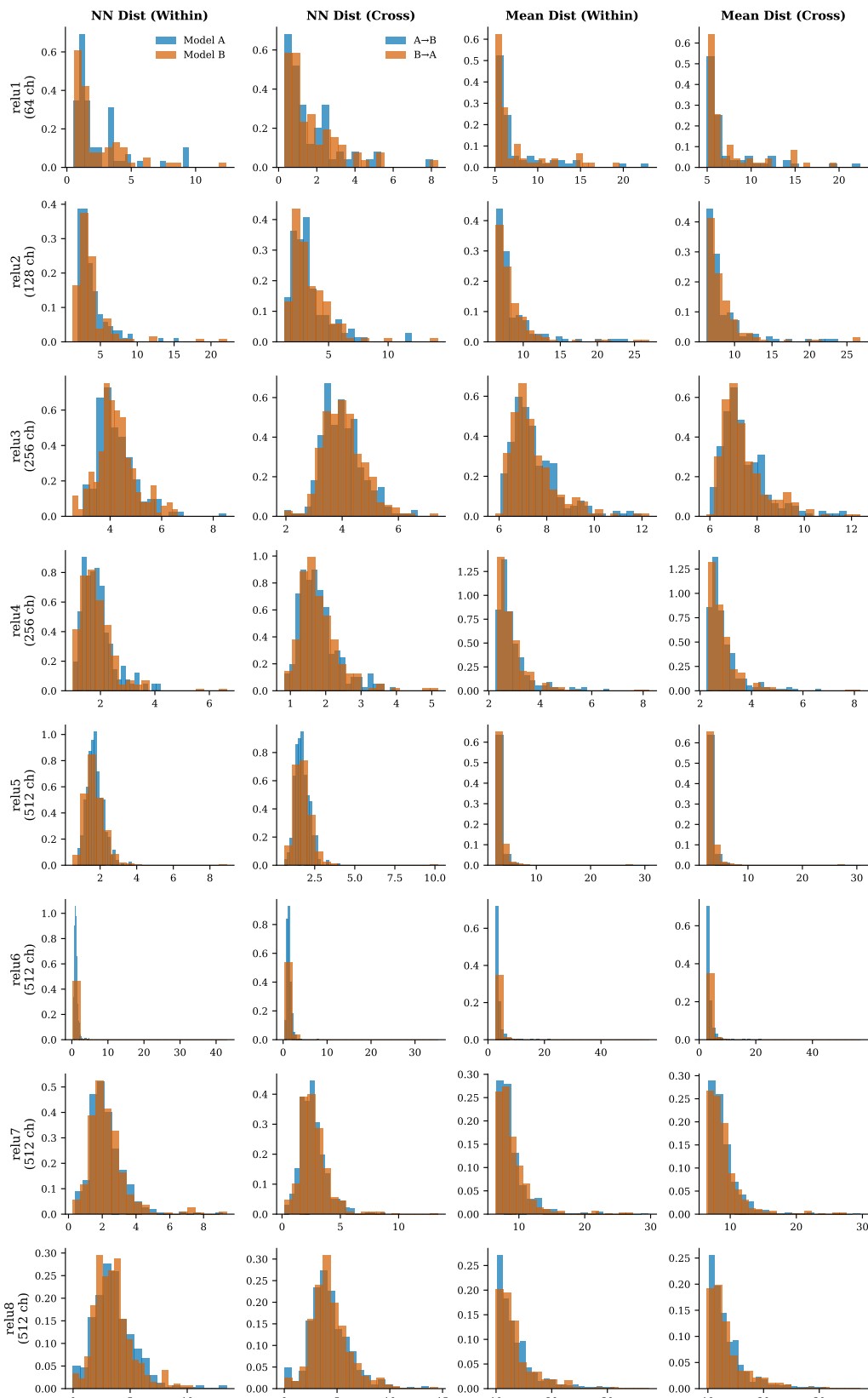

*Figure 17.* Channel distance distributions for two CNNs trained on the same data with different random seeds (as in Table 9). Distances are measured as in Figure 16, but with channel activations instead of neuron activations.

