# OpenReview forum: "Partial Fusion of Neural Networks: Efficient Tradeoffs Between Ensembles and Weight Aggregation"
_ICML.cc/2026/Conference — ICML 2026 regular_

### Official Review · Reviewer_J2A9 · 2026-03-10

**Soundness:** 4
**Presentation:** 3
**Significance:** 4
**Originality:** 4
**Overall Recommendation:** 5
**Confidence:** 3

**Summary:**

In this paper, the authors explore partial fusion of weights as a middle point between ensembling and weights aggregation. With this technique, most similar neurons get fused together, whereas more unique ones are left untouched in the fused network: this results in a wider neural network, but with fewer parameters than the ensemble.

The paper starts with a theoretical exploration of partial weights fusion, with a focus on drawing a bridge between partial fusion and generalised pruning of ensembles. The authors then follow-up with a method to identify neurons to fuse (or prune) based on optimal transport. Finally, the authors test their method empirically for different proportions of neurons being fused, and different interpolation factors.

**Compliance With Llm Reviewing Policy:**

Affirmed.

**Final Justification:**

This paper introduces partial fusion of weights as a middle ground between fusion of weights and ensembling, which is an interesting idea. The main weaknesses raised by the different reviewers revolved around limited empirical validation on more realistic model sizes, lack of quantitative evidence in terms of compute and memory savings, as well as some unclear elements in the paper. Since the authors have rebutted all of these concerns with relevant arguments and new experimental results, the soundness of the revision is better than that of the original submission. I have therefore increased my score and recommend accepting this submission with the suggested revisions.

**Key Questions For Authors:**

**Q1 \- Theoretical exploration:** Like mentioned in my W1, I have some questions regarding the theoretical exploration in this paper. First, the authors write on lines 120-121 column 2: “*in the line of research for layer-by-layer model fusion which we follow*”; could the authors please detail which paper uses a similar setting? The one described by the authors is not the same as in Singh & Jaggi \[1\]. Then, further down in the same paragraph, the authors define their setting as one where representations of Model A and B are equal up to some linear, invertible transformation. Does such an assumption hold in practice, in particular regarding layers of different sizes? Or are matrices $K$ only approximations? If so, the introduction of the notation should be clarified.

**Limitations:**

Yes.

**Strengths And Weaknesses:**

**S1 \- Originality & impact:** I think the idea of partial fusion as a middle ground between weight aggregation and ensembling is very good. It offers good compromises between parameters-efficiency and performance, with results indicating good trade-offs for intermediate values of $\\alpha$.

**S2 \- Experimental evaluation:** The experiments run by the authors are good and in line with the existing literature on the topic \[1\]. They show how partial fusion and generalised pruning work well under multiple proportions of neurons fused/pruned, for both an experiment with specialised vs general knowledge, and a second one with the same training set. The method is evaluated on MNIST with a MLP, and on CIFAR-10 with a VGG-11. Although including an experiment with ResNet would be welcome, I believe the current set of experiments correctly demonstrates the capabilities of the partial fusion method.

**S3 \- Presentation:** The presentation and writing of the paper are good. The Figures are of high quality and help understanding the paper, with Figure 2 providing a clear and intuitive overview of the method.

**W1 \- Theoretical exploration:** Related to my Q1, I have had more difficulties following through the theoretical introduction of partial fusion in Section 2\. I am in particular confused by the assumption that the representations of Model A and B are linked by an invertible linear transformation, which may often not hold in practice especially for unequal layer sizes. If they are just an approximation (as suggested by Section 2.4), the definition should be made clearer. The rest of Section 2 looks good to me.

**W2 \- Efficiency:** I believe that the paper does not discuss the computation cost of partial fusion or generalised pruning. Although Singh & Jaggi \[1\] indicate that solving OT on models such as VGG-11 and ResNet-18 is relatively efficient, seeing it discussed in this paper would be a nice addition, in particular to discuss whether there are differences between the three methods in Figure 5\.

## References

\[1\] Singh, S. P., & Jaggi, M. (2020). Model fusion via optimal transport. *Advances in Neural Information Processing Systems*, *33*, 22045-22055.

---

> ### Author Rebuttal · Authors · 2026-03-30
>
> Thanks a lot for your review and your suggestions for improving our paper. We address all your points raised below:
>
> S2: You mention added experiments with ResNets would be welcome. We address this by performing additional experiments on ResNet-18, showing that the partial fusion method is fully compatible with residual connections and batch norm. See https://anonymous.4open.science/r/pf-33/rn_f.pdf for the results. A more detailed description related to the technical extensions used to achieve this is in our reply to point 1) of reviewer ahME.
>
> W1: We agree the notation regarding the transformations was misleading, and we will add a clearer introduction in this regard. To clarify, we actually do not assume that $K_{\ell}^{A \rightarrow B}$ and $K_{\ell}^{B \rightarrow A}$ are inverses of each other. While they should intuitively be regarded as pseudo-inverses, the presented methods do not require a priori properties of these maps (aside from linearity).
>
> W2: We admit that our current discussion of computation times was quite limited, and our discussion in the revised version will be more detailed. We will differentiate two main aspects, where the first was raised by reviewer wqhJ regarding inference-time performance, see our reply to point 2. of reviewer wqhJ above. The second aspect (which you refer to) is related to the one time cost of computing the parameters of the merged models. As you predicted, using optimal transport for the given model sizes in the paper (the MLPs with 100 hidden dimension and VGG11) is actually very efficient, and solving the respective optimal transport problems never takes more than a second. The fixed-point algorithm (used for Figure 5 a) adds a constant factor related to the number of fixed-point iterations, which is however quite small (it usually converges in less than 10 iterations). Computational times for the partial OT Fusion method are hence always less than a minute overall throughout all cases in the paper. However, the computational times change drastically when considering the generalized pruning (with clustering) method, where clock times are usually tens of hours (even with parallel computing on 48 CPU cores, merging a single model takes 15-30 minutes). We will add more precise statistics in the revised version.
>
> Q1: We hope this is clarified by our response to W1 above: It was indeed simply misleading notation, which we will clarify in the revised version. So our setting is in line with, e.g., Singh and Jaggi (2020) or Ainsworth et al (2023).

---

> > ### Author Rebuttal · Reviewer_J2A9 · 2026-04-02
> >
> > I thank the authors for their rebuttal. I have read through the other reviews, which raise a number of very relevant points, and the corresponding rebuttals. Overall, the quality of the rebuttals is excellent, with few points remaining unaddressed. The additional experiment on ResNet and the related technical adaptations are welcome additions. My opinion is that the method introduced by the authors is novel, interesting, and sufficiently backed by both theory and experiments. I am therefore happy to increase my score from **4: Weak accept** to **5: Accept**.

---

### Official Review · Reviewer_wqhJ · 2026-03-11

**Soundness:** 3
**Presentation:** 3
**Significance:** 2
**Originality:** 3
**Overall Recommendation:** 3
**Confidence:** 3

**Summary:**

This paper introduces partial fusion, a method that interpolates between ensemble inference and weight aggregation like model soup. The idea is to partially align neurons across two networks via partial optimal transport and fuse only the similar neurons, while keeping the remaining units isolated. The resulting model thus retains parts of both networks (ensemble) and averages weights only for matched units (like model soup), aiming to provide a flexible trade-off between model size and predictive performance.

**Compliance With Llm Reviewing Policy:**

Affirmed.

**Final Justification:**

I recognize that this work is interesting and has the potential to make an impact. However, I do not believe that the current experiments and validations are sufficiently strong to support acceptance at ICML this year. I sincerely thank the authors for their response, but my overall assessment remains unchanged, and I therefore maintain my original rating.

**Key Questions For Authors:**

Please refer to the weaknesses above.

**Limitations:**

Please refer to the weaknesses above.

**Strengths And Weaknesses:**

**Strength**

The main concept is clear and well-motivated: it provides a principled continuum between full ensembles and full fusion, rather than treating them as two disjoint choices.


**Weakness**

I have some concerns as follows:

1) Experiments are primarily on MLP + MNIST and VGG11 + CIFAR10. It results in a concern about the applicability of the proposed method to more complex models.

2) The paper claims efficiency mainly in the sense that a modest increase in model size yields disproportionate performance gains when moving from fusion toward ensembles. However, the paper does not provide direct evidence about the efficiency (e.g., wall-clock latency, throughput, memory footprint). I recommend explicitly reporting FLOPs/parameters and actual inference latency for representative settings. Also, I expect that the inference time is not shorten with GPU parallel computing.

3) While  \lambda is a central parameter controlling the interpolation between the two models, the paper mainly sweeps \lambda in plots without providing any analysis.

4) Appendix suggests that pruning can be more parameter-efficient than partial fusion, which raises questions about the practical advantage of the proposed method. The paper also notes that post-finetuning may be needed after merging, especially for CNN. Please compare against a combination of ensembling, pruning, and post-finetuning, and suggest some practical scenario to support the practicality claim.

5) The proposed method is conceptually related to earlier lifelong learning and TIES-Merging in that it partitions neurons into isolated units and mergeable similar units. While the goals differ, discussing how these prior works identify and group similar neurons could help position the contribution more clearly.

In minor, it would be cleaner to start the supplementary material on a new page after the references using \clearpage so that the supplement is visually separated from the reference section.

---

> ### Author Rebuttal · Authors · 2026-03-30
>
> Thanks for your review, we highly appreciate your suggestions for improving the paper. We reply to your points below.
>
> 1\. We agree that applicability to more complex models is crucial, and this was explored too little in the initial submission. We thus include a major extension to ResNets, showing that both residual connections and batch normalization are no obstacles to the applicability of the introduced partial fusion method. See https://anonymous.4open.science/r/pf-33/rn_f.pdf for preliminary results and the reply to point 1) of reviewer ahME for details.
>
> 2\. Thanks for raising these issues. We will add a discussion in the appendix in the revised version. In our reply, we differentiate two regimes: Regime 1 for smaller models, where it is more efficient to consider partially fused models as dense models including zeroes. Regime 2 for larger models, where it is more efficient to consider each non-zero block as a parallel layer.
>
> Storage: Storage size relates practically one-to-one to the number of effective parameters. This is the case even in the first regime by only reconstructing the dense matrix at load time.
>
> Memory: Ensembles are actually reasonably memory efficient, as only the activations of a single model need to be held in memory at a time. This means we mostly cannot claim that partial fusion leads to memory benefits compared to ensembles, as this would only be true for small batch sizes, where the parameter-related memory effects dominate.
>
> FLOPs/Wall-Clock-Time: As we understand it, for a fixed architecture and input size, FLOPs scale linearly with the number of parameters in a network as $\alpha$ varies, and hence in Regime 2, with the number of effective parameters. Therefore, for instance with $\alpha=0.5$ (and assuming equally sized layers), a partially fused network uses 1.75x the FLOPs of a single model, while ensembles use 2x. However, as you pointed out, using Wall-Clock-Time with parallel computing, the effects are more subtle. For small models with small batch size, effects are almost negligible, yet with large batch sizes there are notable benefits compared to ensembles, see https://anonymous.4open.science/r/pf-33/wc_b.pdf. In realistic regimes for large models (like MLP blocks in transformers), Wall-Clock-Time approaches the theoretical FLOP calculation, and Regime 2 with alpha=0.5 gives the expected benefits compared to ensembles, see https://anonymous.4open.science/r/pf-33/wc_w.pdf. To summarize: When implemented suitably, partial fusion provides benefits compared to ensembles regarding computational times, though not always perfectly in line with effective parameters.
>
> 3\. Regarding $\lambda$: We admit the role of $\lambda$ should be discussed more. In the paper, we show benefits for varying $\lambda$ for two reasons: First, benefits on downstream tasks may not always correlate perfectly with performance directly after merging. A new experiment involving fine-tuning should provide insight here: While performance before fine-tuning is often best for $\lambda=0.1$, the performance after fine-tuning is better for $\lambda=0.5$ (see the cases $\alpha=0.2, 0.4$ at https://anonymous.4open.science/r/pf-33/ft_l.pdf). Second, in realistic scenarios a test set to choose $\lambda$ may not be available and thus showing benefits for all $\lambda$ is a more robust indicator of benefits in practice.
>
> 4\. Regarding the practical benefits, we added an experiment involving fine-tuning (see https://anonymous.4open.science/r/pf-33/re_t.pdf), which aims at providing a realistic case in which the introduced methods lead to practically useful aggregated models. Further, regarding pruning of ensembles: We agree this is a strong method and must be taken seriously. However, we emphasize that the parameter efficiency of pruning in relation to partial fusion was perhaps slightly overstated in the initial submission, as the partial fusion method has no way to completely discard unimportant neurons/channels. Thus, a fair comparison regarding parameter count may be to first prune the individual models, and then partially fuse them. With this approach, performance of partial fusion is only slightly affected, yet matches the parameter counts for pruning, see https://anonymous.4open.science/r/pf-33/pr_f.pdf. Further, pruning an ensemble shows limited potential in cases where actual model synergies should be exploited, which can for instance be seen from the results for $\lambda=0.5$ in Figure 10.
>
> 5\. We agree that TIES-Merging is relevant in relation to our work, and our paper can benefit from a more in-depth discussion of the relation. Our current perspective is that TIES-Merging can be regarded as a sophisticated instance of a generalized pruning procedure when applied to the ensemble of just two models. Particularly the treatment of different signs by TIES exploits a symmetry which is not considered by our methods and could be integrated. We will add a short discussion in the revision.

---

> > ### Author Rebuttal · Reviewer_wqhJ · 2026-04-03
> >
> > 1. In the response to reviewer ahME, the authors stated that “Since all reviewers agreed an extension to ResNets is highly beneficial”, but I believe it is inaccurate and clearly overstated. While some reviewers did mention ResNet, I did not agree that an extension from VGG to ResNet is sufficient. In my view, such an extension remains quite limited. For example, architectures involving Multi-Head Self-Attention require substantially more careful consideration, due to factors such as the correlations between queries and keys and the sample-specific adaptive nature of the attention function. In this situation, it becomes much more complex to use the proposed method. Therefore, current validation restricted to conventional CNNs is still not sufficient. At most, it supports the applicability of the method to CNNs.
> >
> > 2. In addition, the current empirical validation remains far below what would be expected under current ICML standards. More comprehensive evaluations are needed, including settings with difficulties closer to model capacity, where pruning may no longer be straightforward. At a minimum, the level of validation should be comparable to that of recent related work. As it stands, the current experimental setup is closer in scope to that used in the 2020 paper (Singh and Jaggi, Model Fusion via Optimal Transport).
> >
> > 3. In the wall-clock time analysis, the authors only vary the MLP width and batch size, and my understanding is that in many cases the proposed method is slower than the ensemble baseline. This again suggests that the method should be tested on substantially more realistic models before strong conclusions can be drawn.
> >
> > 4. The experiments included in (https://anonymous.4open.science/r/pf-33/re_t.pdf) also need to be treated much more rigorously. Adding random neurons is closely related to the literature on network growing, and in this context the results can vary significantly depending on how the random values are initialized or sampled. Accordingly, the authors should demonstrate much more clearly that these experiments have been conducted under a fair and well-controlled setup. The following papers, for example, study closely related issues and show that performance can depend strongly on the sampling strategy.
> >
> > [1] Yuan et al., Accelerated Training via Incrementally Growing Neural Networks Using Variance Transfer and Learning Rate Adaptation
> >
> > [2] Chen et al., Net2Net: Accelerating Learning via Knowledge Transfer
> >
> > [3] Ding et al., Network Expansion for Practical Training Acceleration
> >
> >
> > I appreciate that the authors added experiments on λ and considered discussing related work. However, I still believe that the current extension and validation experiments rely on outdated settings and do not sufficiently demonstrate whether the proposed method is applicable to the models and datasets used in contemporary AI research. For this reason, I will maintain my score.

---

> > > ### Author Response · Authors · 2026-04-04
> > >
> > > Thanks again for taking the time to reply in detail.
> > >
> > > First of all, we're sorry for misrepresenting your view. We did not mean to mislead anyone, though we agree that it ended up as such. On the one hand, we honestly thought you would consider an extension to ResNets beneficial for the paper based on your comment about more complex models. We understand now that this was not what you meant and our phrasing was not correct. On the other hand, the "highly" in front of "beneficial" was overstated; what we meant to convey is that most reviewers listed something in this direction as one of the first points, which we inferred meant it would be very good to include. Again, what we wrote was an incorrectly phrased leap and we should have been more accurate.
> > >
> > > Before replying to your concrete points, we want to more broadly address what we think may be a source of divergence in our exchanges regarding empirical validation/numerical experiments. In this regard, what we want to make clear first is that we don't disagree with your points, we fully see that you are pointing out valid aspects that our paper is lacking. The only thing we want to describe below is our intention regarding what the experiments are meant to achieve (aside from the aspects in which they are lacking).
> > >
> > > **What our experiments don't do**: We acknowledge that our experiments are not designed to demonstrate SOTA performance on modern-scale architectures. We don't want to claim this and we will be very careful that none of our phrasings in the paper make this claim. In particular, we don't want to claim that the presented experiments, also the ones in the rebuttal, are on realistic models or datasets used in practice. In this regard, we realize we may have overstated the finetuning example in our rebuttal. It is meant to illustrate a *type* of task where our method could be practically relevant, not to serve as evidence of practical applicability at scale.
> > >
> > > **What our experiments are supposed to do** is showcase our theoretical and conceptual framework can be transferred to numerics. So our theoretical framework postulates two viewpoints (partial fusion and generalized pruning) to interpolate between model merging and ensembles, and the experiments should simply showcase that the desired characteristics (like model performance is inbetween fully merged models and ensembles) are consistent between theory and numerics. While these experiments are of course only toy-scale, we think that towards the goal of developing scientific understanding related to model merging, such experiments do contain value. For instance, to us, it is a priori fairly surprising that a non-trivial theoretically derived formula like the one of the partial fusion layer (cf.~Figure 4) actually leads to performance as predicted by the theory (in between merged model and ensemble, and sometimes even efficiently in terms of effective parameters), especially across different architectures like MLPs, CNNs and ResNets. Again, the specific models we use are not meant to illustrate modern SOTA architectures, but just generic neural networks architectures with diverse aspects, to exemplify applicability without cherry-picking.
> > >
> > > **Regarding your specific points:**
> > > - Regarding empirical validation: We agree it is important to transfer the introduced methods to more practically relevant settings and large scale modern architectures. The most important extension would of course be transformers in this regard, but as you mentioned, a lot of new complications arise for transformers, and while we view this as a natural next step, we think it goes beyond the scope of the current work.
> > > - We agree that evaluating models close to capacity is valuable. Perhaps our VGG11 experiments already go a bit in this direction, at least judging from Figure 8 in the paper (which shows that even pruning it to 90% hidden dimension leads to a loss of performance around 2.5-5%). So this is a case where our methods are working for a model at capacity (with "at capacity" defined in a particular way, which we hope is not completely different from what you had in mind). We however agree that a more systematic quantitative exploration of this aspect is valuable, and we plan to include an ablation study on the MLP experiments in this regard in the revision.
> > > - Regarding Wall-Clock-Time: What we mainly wanted to illustrate is that the preliminary experiments suggest that for large MLPs with a suitable implementation (Regime 2), the expected benefits as derived from a FLOP calculation can (approximately) be realized. While for small models and large batch sizes there are also some benefits, this is of course less relevant.
> > > - Thanks for pointing out the papers related to random extensions. We will consider the methods therein for the revision and make sure the comparison is clear and fair.

---

### Official Review · Reviewer_c5LZ · 2026-03-13

**Soundness:** 3
**Presentation:** 3
**Significance:** 3
**Originality:** 3
**Overall Recommendation:** 4
**Confidence:** 4

**Summary:**

The paper looks into the middle ground between neural network ensembles and weight aggregation. The main idea is to only fuse the most similar neurons/channels across two models, while leaving the remaining parts isolated, thereby constructing a partially fused network that trades off accuracy against model size.

The authors instantiate this idea via Partial OT Fusion, where the partial alignment is obtained through partial optimal transport, and they additionally introduce a more general viewpoint of generalized pruning, where an ensemble is compressed by either deleting neurons or linearly combining them. Empirically, the paper evaluates these ideas on MLPs on MNIST and VGG11-style CNNs on CIFAR10, both for model aggregation and for single-model pruning, and reports that partial fusion and generalized pruning can often produce favorable parameter/performance tradeoffs relative to pure OT fusion, ensembles, or standard pruning baselines.

**Compliance With Llm Reviewing Policy:**

Affirmed.

**Final Justification:**

I read the rebuttal reply. Most importantly, the additional results on larger-scale models convinced me to increase my rating.

I think this is now a valuable contribution to the ICML research community.

**Key Questions For Authors:**

**Q-1**: The empirical section currently focuses on MNIST MLPs and VGG11 on CIFAR10. Can the authors provide evidence on at least one more modern vision setting, for example, a ResNet- or ViT-style architecture and/or a more challenging dataset? A positive result there would increase my confidence that this is more than a toy-scale proof of concept.

**Q-2**: How much of the observed difference between Partial OT Fusion and clustering is due to the method itself, and how much is due to the feature representation used for matching? Since the appendix states that Partial OT Fusion uses weight-based features while clustering uses activation-based features, an ablation with matched feature types would help clarify this substantially.

**Q-3**: In the same-data CNN setting, the best results seem to rely on keeping several layers fixed as ensembles. Do the authors have a more principled criterion for deciding which layers should remain unfused, beyond the current exploratory study?

**Q-4**: The paper positions itself in the broader model merging literature, but the experiments do not compare against some stronger alignment/merging baselines in the same-architecture setting. Can the authors comment on how Partial OT Fusion relates empirically to such methods, or provide additional comparisons? This would help contextualize the proposed method relative to the current state of the art.

**Limitations:**

yes

**Strengths And Weaknesses:**

**S-1**: I think this is an interesting paper with a novel core intuition. The idea of interpolating between full weight aggregation and full ensembles is natural, but the paper turns this intuition into a reasonably clean formalism. In particular, I like the viewpoint that partial fusion can be interpreted both directly, via partially matched neurons, and indirectly, as a special case of generalized pruning of an ensemble. That gives the paper more conceptual depth than a purely heuristic merging method.

**S-2**: I also found the submission clearly written overall. Figures communicate the method well, and the narrative from standard fusion to partial fusion to generalized pruning is easy to follow. On the empirical side, the split-MNIST experiments show the intended behavior quite clearly: Partial OT Fusion gives a smooth interpolation between OT fusion and ensembles, and the fixed-point variant seems meaningfully better than the simpler matching alternatives. The additional observation that different layers are differently amenable to fusion is also interesting and could be useful for future work.

**W-1**: My main concern is that the empirical case is still too limited. The evaluation is concentrated on MLPs on MNIST and VGG11 on CIFAR10, which makes the paper feel more like a promising proof-of-concept than a convincing demonstration of broad practical relevance. This matters because the paper is framed quite broadly around neural network fusion and compression, yet the experiments do not show behavior on more modern computer vision architectures or more challenging datasets.

**W-2**: Related to this, the strongest same-data CNN result appears to rely on manually keeping several layers fixed as ensembles, which is interesting, but also suggests that the plain method is not yet robust enough in realistic fusion settings. The generalized pruning results are also somewhat mixed: clustering often performs better, but at the cost of solving an expensive clustering problem, and the appendix makes clear that the optimization procedure itself is nontrivial and can require many restarts. Thus, the “efficient tradeoff” in the title seems to refer mainly to parameter/accuracy tradeoffs, not necessarily computational efficiency of the method itself.

**W-3**: Another weakness is that some of the comparisons are not yet fully convincing. The main baselines are OT Fusion, unstructured pruning, and the proposed clustering variant, which are reasonable choices, but I missed stronger comparisons to more recent model merging/alignment methods in the same-architecture setting. In addition, the comparison between Partial OT Fusion and clustering is not entirely apples-to-apples, since the appendix states that Partial OT Fusion uses weight-based features while clustering always uses activation-based features. That choice may be perfectly reasonable, but it makes it harder to separate method effects from feature-design effects. I would also love to see some discussion about the difference of this approach to weight space learning, a paradigm that learns from weights of multiple models to generate weights for one model:

-->
*A Survey of Weight Space Learning: Understanding, Representation, and Generation*,
Xiaolong Han, Zehong Wang, Bo Zhao, Binchi Zhang, Jundong Li, Damian Borth, Rose Yu, Haggai Maron, Yanfang Ye, Lu Yin, Ferrante Neri
https://arxiv.org/abs/2603.10090, 2026

**Conclusion**: More generally, the technical contribution is sound but somewhat modest: the theory is mostly a formalization of the construction and its relation to transport/clustering, rather than a deeper explanation of when partial fusion should work or guarantees on the resulting fused model quality. Overall, I find the paper technically solid and conceptually interesting, but the current evidence is not yet broad enough for me to be fully convinced of its significance.

---

> ### Author Rebuttal · Authors · 2026-03-30
>
> Thanks a lot for your detailed feedback and also your assessment regarding the strengths and weaknesses of our paper. We respond below to your main concerns, focusing on your concrete questions, which we hope address the weak points you raised simultaneously.
>
> Q-1 (and W-1): We agree with the concern that the currently covered architectures and the overall empirical case are too limited. In the initial submission, we chose to focus purely on the conceptual idea, but we agree that this leaves the broad framing that we're aiming for like a vague promise, and not something scientifically sound. To address this, we now include two main additions: First, we provide an extension to ResNets, showing that both residual connections and batch normalization are no obstacles to the applicability of our partial fusion method. See https://anonymous.4open.science/r/pf-33/rn_f.pdf for preliminary results and the reply to point 1) of reviewer ahME regarding a short explanation of the key technical extensions. Second, we include a clean study involving fine-tuning, which robustly shows that partial fusion leads to practical gains in a realistic scenario, both for MLPs and ResNets. See https://anonymous.4open.science/r/pf-33/re_t.pdf for the results.
>
> Q-2: Regarding the fairness in terms of features used in the comparison between partial OT Fusion and clustering, our initial story was meant as follows: In Figure 1 we first compare Partial OT Fusion with different features, then choose the best version to compare against clustering. As clustering "wins" against the best version for partial OT Fusion (at the one-time-compute-cost of solving the costly clustering problem), we don't need to look at other features for clustering. However, we realize now that this story is unnecessarily convoluted, and the reader would benefit much more from a clean comparison between the methods each using the same features (particularly for the CNN case as well), and thus we will simply include it in the revised version. As the clustering method takes very long to compute, we could not provide the results for this rebuttal though.
>
> Q-3: We believe there are two principled criteria for choosing which layers to fuse, which we agree should be elaborated on more in the paper. However, we also point out that we do not rigorously test them, so take those as hypotheses rather than facts. The first principle comes from Git Re-Basin (see Figure 4 in Ainsworth et al, 2023) who establish that wider networks are easier to merge. We think this can be extrapolated to mean that wider layers within a given network are easier to merge. The second criterion is error propagation: Changing earlier layers leads to errors which propagate through to later layers, while changing later layers leads to more local errors, hence they are more forgiving for fusion. Our exploratory experiments are consistent with both principles (though of course not a rigorous test thereof), as we see that fixing earlier layers, which happen to be narrower in VGG11, leads to disproportionate improvements.
>
> Q-4: Regarding the context of our work related to existing methods: We first want to emphasize that the main goal of our paper is not to compete with existing model merging methods, but rather to establish synergy by showcasing how a given method may be extended by providing the option of interpolating it with ensembles. We agree however that more synergy should be established and we will include additional results related to Git Re-Basin (as a stronger baseline method in the same architecture setting, see Ainsworth et al, 2023) and ZipIt (as a method strongly related to the generalized pruning procedure studied in the paper, see Stoica et al, 2024) in a revision (as also addressed in the reply to the key questions by reviewer ahME). As all our experiments are currently focused on the case of independently trained models, we believe a comparison with methods focusing on fine-tuned models (like TIES-Merging or Task Arithmetic) would go beyond the scope of the current work.
>
> We also appreciate the reference to the survey on Weight Space Learning, which we were unaware of. The way we see it, our work mainly relates to the understanding part of the Weight Space Learning paradigm. Our intuition is that weight aggregation methods basically define what "a line connecting two models" means in this space. Our work extends this slightly by exploring the geometry of the triangle of two models and their ensemble. We will reference the survey and discuss this relation in the revision.
>
> Finally, in W-2 you raise concerns regarding a more precise treatment of computational efficiency: We agree that more must be said, see our response to point 2. by reviewer wqhJ regarding computational aspects related to inference, and our respose to point W2 of reviewer J2A9 regarding computational cost of computing the merged models.

---

> > ### Author Rebuttal · Reviewer_c5LZ · 2026-04-04
> >
> > I would like to thank the authors for their reply. However, I am unsure about this rebuttal.
> >
> > Having read the **ICML 2026 Author Instructions** from https://icml.cc/Conferences/2026/AuthorInstructions, I am wondering if this rebuttal is breaking the official ICML rebuttal guidelines that state:
> >
> > "As reviewing is double-blind, the response should not contain information that could reveal the authors’ identities. In addition, the response **should not contain non-anonymized URLs**, URLs for personal websites, or “shortened” URLs (e.g., as provided via tinyurl, which could log a reviewer’s IP). Reviewers are not expected to follow external URLs in the response."
> >
> > The provided rebuttal (and the rebuttal to other reviewers) has included two external links, and I would like to ask the AC to guide me in my response to this rebuttal.

---

> > > ### Author Response · Authors · 2026-04-04
> > >
> > > Thanks for acknowledging our reply and raising this concern.
> > >
> > > While we understand that you want a more official reply from the AC, we can also offer our point of view: We tried to follow the guidelines communicated in an email from the 24th of March, stating in one point:
> > >
> > > > Anonymity and Links: Your responses to reviewers should not contain or link to any identifying information that may violate the double-blind reviewing policy. While links are allowed, reviewers are not required to follow them, and links may only be used for figures (including tables) and captions that describe the figure (no additional text). All links must be anonymous to preserve double-blind review, both in the URL and the destination.
> > >
> > > As far as we know, everything we linked to is anonymous and within the rules (just figures and tables with captions). We included the links as optional supporting evidence, but the rebuttal is intended to be self-contained without them.

---

### Official Review · Reviewer_ahME · 2026-03-13

**Soundness:** 2
**Presentation:** 3
**Significance:** 3
**Originality:** 3
**Overall Recommendation:** 5
**Confidence:** 2

**Summary:**

The paper extends existing OT-based fusion and neuron-matching methods to a partial-fusion setting, allowing model merging to achieve a more flexible tradeoff between storage cost and predictive performance based on neuron similarity. It further places this procedure under a unified generalized-pruning perspective.

**Compliance With Llm Reviewing Policy:**

Affirmed.

**Final Justification:**

My main concerns in the initial review were the relatively limited experimental scale, the lack of validation on more modern architectures and more realistic settings, and the moderate originality of the method relative to prior work on model fusion and neuron matching. In the rebuttal, the authors addressed my main empirical concerns fairly well, which made my overall evaluation more positive.

**Key Questions For Authors:**

Could the authors include comparisons with stronger baselines and validate the method on larger-scale datasets and models?
Given that the proposed method is already relatively sophisticated, the current experiments, which are mainly based on simpler baselines and relatively small-scale settings, are not yet sufficient to fully demonstrate its unique gains. It would be helpful to compare against more advanced fusion, neuron-matching, or pruning methods, and to further evaluate the approach on larger datasets and more modern architectures. Such additional experiments would make it clearer whether the observed improvements truly arise from the core idea of partial fusion itself, or whether they are driven more by the added algorithmic complexity and method design.

**Limitations:**

Yes

**Strengths And Weaknesses:**

Strengths

1. The core idea is clear and provides a unified perspective. The main intuition of the paper is quite straightforward: only fuse the subset of neurons that are most similar across the two models, while keeping the remaining ones as independent units. This creates a more flexible performance–cost tradeoff between full ensemble and complete fusion. In addition, the paper introduces the perspective of generalized pruning, which connects model fusion with model compression and gives the work a more unified conceptual framing.

2. The method is formulated in an elegant way. The paper uses OT / partial OT to model partial matching between neurons, and unifies standard fusion, partial fusion, and ensemble into a continuous parameterized family. This mathematical formulation is natural and helps clearly explain the relationship among these different methods.

3. The experimental results generally support the main claim of the paper. The experiments support the claim that partial fusion can provide a more flexible tradeoff between model size and performance than full fusion, and in several settings can approach ensemble performance while retaining lower computational cost.

Weaknesses

1. The experimental scale is still relatively limited, which weakens the empirical support. The main experimental settings are largely concentrated on: MLPs on MNIST, VGG11 / small CNNs on CIFAR-10, relatively toy-style settings involving split datasets or same-dataset scenarios. Since the proposed method is already fairly sophisticated, I think stronger empirical validation on more complex datasets and more modern architectures would be needed to fully demonstrate its practical value and persuasiveness. At present, these smaller-scale experiments mainly show that the method is feasible in toy settings, but are not yet sufficient to convincingly support its effectiveness in more realistic model fusion or compression scenarios.

2. The level of novelty is somewhat limited. Although the paper offers a unified view through partial fusion and generalized pruning, at its core the method is closer to constructing an intermediate interpolation between full fusion and ensemble, fusing only a subset of similar neurons while leaving the rest independent. This is a reasonable and useful idea, but it feels more like a natural extension of existing neuron matching, OT-based fusion, and pruning ideas, rather than a particularly strong new theoretical contribution or a fundamentally new phenomenon.

---

> ### Author Rebuttal · Authors · 2026-03-30
>
> Thanks a lot for reviewing and providing your perspective and suggestions. We respond below to your main points raised:
>
> 1\) The main weakness you raise (shared by all other reviewers) was to present a stronger experimental scale and more realistic empirical validations. We provide two main new experiments to address this:
>
> 1.1) ResNets: We show that the basic partial fusion mechanism works as well for ResNets, so residual connections and batch norm are no obstacles to apply the ideas from the paper and that in such settings one can similarly interpolate between weight aggregation and ensembles. See https://anonymous.4open.science/r/pf-33/rn_f.pdf for preliminary results.
>
> Technical details on ResNet extension: The basic aspects of ResNets are treated similarly to OT-Fusion (Singh & Jaggi, 2020), but with some key adjustments: First, we additionally constrain transport maps to be equal for layers that are connected by an identity skip connection, consistent with the fact that less permutation invariance is present in ResNets. Second, we employ the fixed-point iteration with global weight matching. Finally, to obtain the running statistics of batch norm for the (partially) fused models, we use the recalibration method as described in REPAIR and Git Re-Basin (Jordan et al, 2023 and Ainsworth et al, 2023), or if insufficient data for recalibration is available, we also introduce a new method to compute running statistics in a completely data-free way based on estimating the covariance of neurons through the cosine similarity of incoming weight matrices (used in the new experiment in point 1.2 below).
> We also briefly point out that working out the extension to ResNets was definitely not trivial, and that we had already invested considerable effort into it alongside our initial submission, but did not finish it by the deadline. Since all reviewers agreed an extension to ResNets is highly beneficial for the main points in the paper (which we agree with), we prioritized the completion for this rebuttal.
>
> 1.2) Confidential data with finetuning: We present an experiment including fine-tuning, both for MLPs and for ResNets, which shows that the studied method can really lead to practically important improvements on a task. The basic premise is that two models are trained on separate confidential data and afterwards fine-tuned on a small public dataset. We show that partial fusion leads to efficient trade-offs in this regard, see https://anonymous.4open.science/r/pf-33/re_t.pdf. Importantly, we show that the limiting factor is really the efficiently shared information through partially fusing networks, and not just using larger models.
>
> 2\) Regarding the level of novelty: We agree that at its core, our theoretical work mainly provides a unified perspective on (partial) model merging and pruning. We tried to contribute by working it out as cleanly as possible. What we consider the main theoretical novelty is making the presented ideas work with non-trivial aspects like the fixed point iteration presented in Section 2.5, or (now with the added ResNet experiments) making aspects like Batch Norm partially fuse-able as well.
>
> Regarding your key questions:
> While we hope that point 1) above addresses the part of the question related to more modern architectures and its unique gains in practically important tasks, we further want to address the point raised regarding stronger baselines:
> First, let us emphasize again that our main goal with the paper is not to compete with existing fusion or neuron-matching methods, but rather to supplement them with the option of interpolating them with ensembles. However, we agree that this synergy is so far only explored in detail in one test case, and outlining this synergy with more advanced methods would benefit the paper. To this end, we will address in the appendix how our method can similarly be combined with approaches like Git Re-Basin (Ainsworth et al, 2023); in fact, any permutation-based approach can be used as the weight aggregation baseline which can be combined with the partial fusion method. Similarly, as already hinted at in Section 2.3, ZipIt! (Stoica et al, 2024) relates strongly to the class of generalized pruning methods, and we will more clearly compare this method to our clustering approach from Section 2.4 in the revision. Finally, let us mention that methods aimed at fine-tuned models sharing an initialization (like TIES-Merging or Task Arithmetic) are incompatible with the experimental settings we consider (where we use independently trained models), and thus we consider a precise exploration of connections with these methods beyond the scope of the current paper.

---

> > ### Author Rebuttal · Reviewer_ahME · 2026-04-01
> >
> > Thank you for the rebuttal. The authors have addressed all of my questions and provided sufficient clarification on my main concerns. Based on the response, I am revising my score to 5.

---

### Decision · Program_Chairs · 2026-04-30

**Decision:**

Accept (regular)

**Comment:**

The submission explores the middle ground between neural network ensembles and weight aggregation by introducing a framework for "partial fusion". The presented method interpolates between these two extremes by aggregating only the weights of neurons exhibiting high similarity (using partial optimal transport to identify these matrices) while leaving dissimilar neurons isolated. The approach is contextualised through a generalised pruning perspective, where an ensemble is compressed by either deleting neurons or linearly combining them based on similarity.

Reviewers appreciated the novel core intuition of the work, as well as the clearly presented motivation for creating a principled continuum between full ensembles and weight fusion. The use of partial optimal transport to model neuron matching was described as elegant and natural, with reviewers admiring the reasonably clean formalism unifying standard fusion, partial fusion and ensembles into a continuous parameterised family.

Opinions were more mixed regarding the experimental validation, however. While Reviewers ahME, c5LZ, and J2A9 were satisfied and increased their scores, Reviewer wqhJ maintained that the ResNet-18 validation study remained below the expectations of the field, as the models of interest increasingly shift toward transformer architectures. I agree that the initial submission is highly undermined by the lack of larger model architectures in the experiments, and the experiments involving ResNet-18 go a long way toward addressing this deficiency. Transformers appear to be a bridge too far, but a more thorough treatment of deep neural networks seems warranted. Discussions about FLOP improvements were also welcome to see in the rebuttal, as the practical motivation is quite limited.

The reviewers acknowledge that all of the concerns have been raised in the rebuttal period, but there are many critical modifications to be made here to expand the value of the submission beyond a niche theoretical audience, and I expect upcoming revisions to be substantial. I strongly recommend that the authors take the feedback on board and make the necessary revisions to inspire practical audiences.